# Human-in-the-Loop Optimization for Deep Stimulus Encoding in Visual Prostheses

**Jacob Granley**
Department of Computer Science
University of California, Santa Barbara
`jgranley@ucsb.edu`

**Tristan Fauvel**
Institut de la Vision, Sorbonne Université
17 rue Moreau, F-75012 Paris, France
Now with Quinten Health
`t.fauvel@quinten-health.com`

**Matthew Chalk**
Institut de la Vision, Sorbonne Université
17 rue Moreau, F-75012 Paris, France
`matthew.chalk@inserm.fr`

**Michael Beyeler**
Department of Computer Science
Department of Psychological & Brain Sciences
University of California, Santa Barbara
`mbeyeler@ucsb.edu`

## Abstract

Neuroprostheses show potential in restoring lost sensory function and enhancing human capabilities, but the sensations produced by current devices often seem unnatural or distorted. Exact placement of implants and differences in individual perception lead to significant variations in stimulus response, making personalized stimulus optimization a key challenge. Bayesian optimization could be used to optimize patient-specific stimulation parameters with limited noisy observations, but is not feasible for high-dimensional stimuli. Alternatively, deep learning models can optimize stimulus encoding strategies, but typically assume perfect knowledge of patient-specific variations. Here we propose a novel, practically feasible approach that overcomes both of these fundamental limitations. First, a deep encoder network is trained to produce optimal stimuli for any individual patient by inverting a forward model mapping electrical stimuli to visual percepts. Second, a preferential Bayesian optimization strategy utilizes this encoder to optimize patient-specific parameters for a new patient, using a minimal number of pairwise comparisons between candidate stimuli. We demonstrate the viability of this approach on a novel, state-of-the-art visual prosthesis model. We show that our approach quickly learns a personalized stimulus encoder, leads to dramatic improvements in the quality of restored vision, and is robust to noisy patient feedback and misspecifications in the underlying forward model. Overall, our results suggest that combining the strengths of deep learning and Bayesian optimization could significantly improve the perceptual experience of patients fitted with visual prostheses and may prove a viable solution for a range of neuroprosthetic technologies.

## 1 Introduction

Sensory neuroprostheses are devices designed to restore or enhance perception in individuals with sensory deficits. They often interface with the nervous system by electrically stimulating neural tissue in order to provide artificial sensory feedback to the user [1, 2]. For instance, visual prostheses have the potential to restore vision to people living with incurable blindness by bypassing damaged parts of the visual system and directly stimulating the remaining cells in order to evoke visual percepts (phosphenes) [3–6]. However, patient outcomes with current technologies are limited. Patients

37th Conference on Neural Information Processing Systems (NeurIPS 2023).

Figure 1: *Left*: Deep stimulus encoder (DSE). A forward model ($f$) is used to approximate the perceptual response to electrical stimuli, subject to patient-specific parameters $\phi$. An encoder ($f^{-1}$) is then learned to minimize the perceptual error between predicted and target percept. *Right*: Human-in-the-loop optimization (HILO). Patient-specific parameters $\phi$ of the DSE are optimized with user preferences: the patient performs a series of binary comparisons between percepts evoked with different encoders. New pairs of parameters to compare are adaptively selected so as to efficiently find the parameters maximizing the patient's preference. The target changes each iteration.

require extensive training to learn to interpret the evoked percepts, which are typically described as "fundamentally different" from natural vision [7]. Moreover, phosphene appearance varies widely across patients [8], making personalized stimulus optimization a major outstanding challenge [9].

Much work has gone into developing computational models that can predict the neuronal or perceptual response to an electrical stimulus [8, 10, 11] (often called forward models). Once the forward model is known, a deep neural network can approximate its inverse, thereby identifying the required stimulus to elicit a desired percept [12–14]. However, these inverse models typically assume perfect knowledge of the patient-specific mapping from stimulation to perception (which is not practically feasible) and are heavily reliant on the forward model's accuracy over the entire stimulus space.

Alternatively, Bayesian optimization has been successful in personalizing stimulation strategies for many existing neural interfaces [15, 16]. However, this approach is limited in practice because it requires the stimulus dimension to be small (typically $< 30$ [17], which is orders of magnitudes smaller than the number of stimulus parameters in current implants), and optimization must be repeated for every stimulus. Moreover, visual prosthesis users can typically only give indirect feedback (e.g., verbal phosphene descriptions), unsuitable for traditional Bayesian optimization.

To address these challenges, we propose a novel framework that integrates deep learning-based stimulus inversion into a preferential Bayesian optimization strategy to learn a patient-specific stimulus encoder (Fig. 1). First, a deep stimulus encoder (DSE) is trained to optimize stimuli assuming perfect knowledge of a set of patient-specific parameters (Fig. 1, *left*). Second, we embed the DSE within a human-in-the-loop optimization (HILO) strategy based on preferential Bayesian optimization, which iteratively learns the ground-truth patient-specific parameters through a series of 'duels', where the patient is repeatedly asked their preference between two candidate stimuli. The resulting DSE can then be deployed as a personalized stimulation strategy.

To this end, we make the following contributions:

- We introduce a forward model for retinal implants that achieves state-of-the-art response predictions. Unlike previous models, this allows us to train a deep stimulus encoder to predict optimal stimuli across 13 dimensions of patient-specific parameters.

- We propose a personalized stimulus optimization strategy for visual prostheses, where a human-in-the-loop optimization (HILO) Bayesian optimization algorithm iteratively learns the optimal patient-specific parameters for a deep stimulus encoder.

- We demonstrate the viability of our approach by conducting a comprehensive series of evaluations on a population of simulated patients. We show HILO quickly learns a personalized stimulus encoder and leads to dramatic improvements in the quality of restored vision, outperforming existing encoding strategies. Importantly, HILO is resilient to noise in patient feedback and performs well even when the forward model is misspecified. Code for the forward model, DSE, and HILO algorithm is available at `https://github.com/bionicvisionlab/2023-NeurIPS-HILO`.

## 2   Background and Related Work

**Visual Neuroprostheses**   Numerous groups worldwide are pursuing a visual prosthesis that stimulates viable neuronal tissue in the hope of restoring a rudimentary form of vision to people who are blind (Fig. 2, *left*) [3–6]. Analogous to cochlear implants, these devices electrically stimulate surviving cells in the visual pathway to evoke visual percepts (phosphenes). Existing devices generally provide an improved ability to localize high-contrast objects and perform basic mobility tasks.

Much work has focused on characterizing phosphene appearance as a function of stimulus and neuroanatomical parameters [2, 10, 18–21]. In epiretinal implants, phosphenes often appear distorted due to inadvertent activation of nerve fiber bundles in the optic fiber layer of the retina [8], causing elongated percepts (Fig. 2, *center*). In addition, the exact brightness and shape of these elicited percepts depends on the applied stimulus [19] and differs widely across patients (Fig. 2, *right*). Granley *et al.* [11] captured these individual differences by including a set of patient-specific parameters in their phosphene model, denoted by $\phi$, which includes both neuroanatomical (e.g., implant location) and stimulus-related parameters (e.g., how brightness scales with amplitude).

**Deep Stimulus Encoding**   Many works attempt to mitigate distortions in prosthetic vision, but do not describe comprehensive stimulation strategies [22–24]. Those that describe strategies in detail typically require simplification [25] or strong assumptions [26] to be used in practice. Due to the complexities of optimization, deep learning-based stimulus encoders have risen in popularity [12–14]. In [12], authors proposed an innovative approach where the latent representations of an autoencoder are treated as stimuli and decoded with a phosphene model. However, they used an unrealistic binary phosphene model. Their approach has since been adapted for cortical models [27], and for non-differentiable forward models [13]. Granley *et al.* [14] generalized the approach, showing it could work with realistic forward models across a small range of patients without needing to retrain.

Given a forward (phosphene) model $f$ (mapping stimuli to percepts given $\phi$), it is straightforward to show that the optimal stimulus encoder (mapping target images to stimuli) is the pseudoinverse of $f$ [14]. However, to account for the wide range of individual differences in phosphene perception, most realistic forward models are highly nonlinear and not analytically invertible. Thus, previous works have proposed to use the forward model [12, 14] as a fixed decoder within a deep autoencoder trained to minimize the reconstruction error between target images and the predicted percepts. After training, the encoder can be extracted and used to encode target visual inputs in real time. Deep

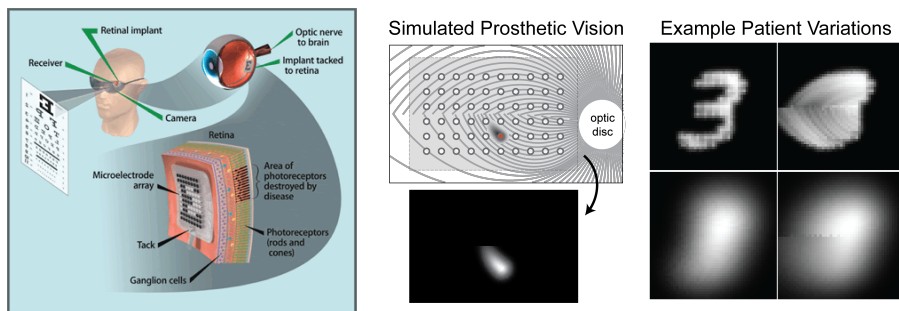

Figure 2: *Left*: Visual prosthesis. Incoming target images are transmitted from a camera to an implant in the retina, which encodes the image as an electrical stimulus pattern. *Center*: Electrical stimulation (red disc) of a nerve fiber bundle (gray lines) leads to elongated tissue activation (gray shaded region) and a phosphene (bottom). *Right*: The same stimulus parameters may lead to widely varying visual perceptions in different patients. Adapted with permission from [14].

stimulus encoders trained using this approach produce high quality stimuli, but assume knowledge of $\phi$. Additionally, if the forward model $f$ is not extremely accurate over the whole stimulus space, then the encoder network might learn to exploit inaccuracies in the model, producing stimuli that don't generalize to real patients [14]. We utilize an enhanced variant of this approach in our experiments.

**Preferential Bayesian Optimization** Preferential Bayesian optimization (PBO) is an efficient method for optimizing expensive black-box functions based on binary comparisons [28, 29]. Since the subject's response to stimulation cannot be directly observed, PBO instead builds a Bayesian model of the subject's preferences, $g$, typically modeled using a Gaussian process. An approximate inference algorithm (expectation propagation; [30, 31]) is used to infer the posterior distribution of the preference function given binary comparison data, $p(g|\mathscr{D})$, which is then used to select new configurations for the next trial according to an acquisition rule. The acquisition rule must balance the exploration-exploitation trade-off inherent to any black-box optimization problem [32].

PBO was previously used to tune BCI stimulation parameters for transcranial [33] and spinal cord stimulation [34]. However, these works directly optimized only a handful of stimulation parameters and cannot translate to visual prostheses, where complex and varying visual inputs have to be mapped to high-dimensional stimuli. To this end, Fauvel & Chalk [35] reduced optimization complexity by inverting a model of phosphene perception, then used PBO to generate encodings preferred by sighted subjects viewing simulated prosthetic vision. However, a linear approximation was used to invert the perception model, which is unrealistic for real-world applications.

**Summary** We identify 3 main limitations of previous work that this study aims to address:

- **Generalizability of deep stimulus encoders.** Autoencoder-like deep stimulus encoders can accurately optimize stimuli, but require perfect knowledge of patient-specific parameters [14], which can be difficult or impossible to determine in practice [8, 11]. Further, these approaches heavily rely on the accuracy of the forward model [13, 14], while real patients will likely deviate from the forward model. We overcome this limitation by optimizing the learned stimulus encoder based on patients' preferences, which we show is not bounded by a misspecified forward model.
- **Applicability of Bayesian optimization.** Bayesian optimization is ideally suited for optimizing stimulation parameters based on limited, noisy measurements, but can only optimize a small number of parameters [17]. We train a deep stimulus encoder to output optimal stimuli for any specific patient with known parameters, reducing the search space from the entire stimulus space (hundreds or thousands of parameters) to the low-dimensional space of patient-specific parameters (13 parameters with our model), enabling Bayesian optimization.
- **Simplistic models of perception.** Most previous approaches use overly simplified forward models that do not match empirical data [8, 19]. More accurate models [11] are too computationally expensive to support deep stimulus optimization over a wide range of patients. We overcome this by developing a new phosphene model that matches patient data better than existing models, with significantly reduced memory and time complexity

## 3   Methods

**General Framework** We consider a system attempting to optimize stimuli for a new patient, specified by a set of (unknown) parameters $\phi$. The goal of optimization is a patient-specific stimulus encoder mapping target perceptual responses $\mathbf{t}$ (e.g., visual percepts) to stimuli $\mathbf{s}$.

We assume there exists a forward model $f$ which predicts the patient's perceptual response to stimulation: $\hat{\mathbf{t}} = f(\mathbf{s}; \phi)$. It follows that the optimal stimulus encoder is the (pseudo)inverse of $f$. The inverse can be approximated using an autoencoder-like deep neural network, with $f$ as the fixed decoder and $\hat{f}^{-1}$ as the learned stimulus encoder [14]. The encoder's weights $w$ are updated using gradient descent to minimize the reconstruction error, measured by some distance function $d$, between the target $\mathbf{t}$ and the predicted response $\hat{\mathbf{t}}$ averaged across patients and a dataset of targets (Eq. 1):

$$w \approx \underset{w}{\arg\min} \; \underset{\mathbf{t}, \phi}{\mathbb{E}} \left[ d(\mathbf{t}, \hat{\mathbf{t}}) \right] \tag{1}$$

$$\hat{\mathbf{t}} = f(\hat{f}_w^{-1}(\mathbf{t}, \phi); \phi) \tag{2}$$

Table 1: Patient-Specific Parameters $\phi$

| | Phosphene Model Parameters | | | | | | | | | | Implant Parameters | | |
|---|---|---|---|---|---|---|---|---|---|---|---|---|---|
| | $\rho$ (dva) | $\lambda$ | $\omega$ | $a_0$ | $a_1$ | $a_2$ | $a_3$ | $a_4$ | $OD_x$ ($\mu$m) | $OD_y$ ($\mu$m) | x ($\mu$m) | y ($\mu$m) | rot (deg) |
| Lower | 1.5 | .45 | .9 | .27 | .42 | .005 | .2 | -0.5 | 3700 | 0 | -500 | -500 | -30 |
| Upper | 8 | .98 | 1.1 | .57 | .62 | .025 | .7 | -0.1 | 4700 | 1000 | 500 | 500 | 30 |

Once trained, the encoder can accurately predict stimuli, but requires knowledge of the patient-specific parameters $\phi$. For a new patient, Bayesian optimization is used to optimize $\phi$ based on user feedback, thereby learning a personalized DSE. Since the patient's response cannot be directly measured for visual prostheses, the user is presented with a 'duel', i.e. a binary comparison, where they are asked to decide which of two candidate stimuli they prefer [35]. A Gaussian process model of preferences is updated based on the patient's response, and an acquisition function is used to generate new candidate stimuli. The process can be repeated to iteratively tune the DSE to the patient's preferences.

**Phosphene Model**  The phosphene model is a differentiable approximation of the underlying biological system (also called a forward model [14]), which maps an electrical stimulus to a visual percept. Although phosphene models exist for visual prostheses, current models either do not match patient data well [8, 10, 19], or are too computationally expensive [11].

Thus, we developed a new phosphene model for epiretinal prostheses (See Appendix A.1 for full details). The model takes in a stimulus vector $\mathbf{s} \in \mathbb{R}^{n_e \times 3}$ specifying the frequency, amplitude, and pulse duration of a biphasic pulse train on each of $n_e$ electrodes. The output phosphene for each electrode is a Gaussian blob centered over the electrode's location $\mu_e(\phi)$ with covariance matrix $\Sigma_\mathbf{e}(\mathbf{s}, \phi)$ constructed so that the resulting percept will have area $\rho_e(\mathbf{s}, \phi)$, eccentricity $\lambda_e(\mathbf{s}, \phi)$ and orientation $\theta_e(\phi)$. These functions allow phosphene properties to vary locally with stimulus (e.g., current spread) and anatomical parameters (e.g., electrode location, underlying axon nerve fiber bundle trajectory). The percept for an electrode located at $\mu_e$, is a multivariate Gaussian blob, renormalized to have maximum brightness $b_e(\mathbf{s}, \phi)$:

$$b(x,y) = 2\pi b_e \det(\mathbf{\Sigma_e}) \, \mathcal{N}([x,y]^\top | \mu_e, \mathbf{\Sigma_e}), \tag{3}$$

where $b_e$, $\mu_e$, and $\Sigma_e$ are implicitly parametrized by $\mathbf{s}$ and $\phi$. The covariance matrix $\mathbf{\Sigma_e} = \mathbf{R}\mathbf{\Sigma_0}\mathbf{R}^T$ is calculated from the eigenvalue matrix $\mathbf{\Sigma_0}$ and a rotation matrix $\mathbf{R}$:

$$\Sigma_0 = \begin{bmatrix} \alpha_e^2 & 0 \\ 0 & \beta_e^2 \end{bmatrix}, \qquad R = \begin{bmatrix} \cos\theta_e & -\sin\theta_e \\ \sin\theta_e & \cos\theta_e \end{bmatrix}.$$

The eigenvalues $\alpha_e$ and $\beta_e$ depend on the intended phosphene area ($\rho_e$) and elongation ($\lambda_e$):

$$\alpha_e^2 = -\frac{\rho_e \sqrt{1-\lambda_e^2}}{2\pi}, \qquad \beta_e^2 = -\frac{\rho_e}{2\pi\sqrt{1-\lambda_e^2}}.$$

Blobs from individual electrodes are summed into a global percept [36]. Although the sum across electrodes is linear, modulating the size and eccentricity of phosphenes with stimulus parameters makes the final result a nonlinear function of stimulus parameters, preventing analytic inversion. Motivated by previous studies, we used a square $15 \times 15$ array of $150\mu$m electrodes, spaced $400\mu$m apart [14]. In total, the model is parameterized by 13 patient specific parameters, shown in Table 1. The ranges for each parameter were chosen to conservatively encompass all observed patients, centered on the mean value across patients [8, 10, 11, 19, 37].

**Deep Stimulus Inversion**  A deep stimulus encoder (DSE) is a deep neural network responsible for approximately inverting the forward model to produce the optimized stimulus for a target image and a specific patient ($\mathbf{s}_\phi = \hat{f}^{-1}(\mathbf{t}, \phi)$) [14]. We used a network (45M parameters) consisting of blocks, each containing 3 fully connected layers, batch normalization, and a residual connection. The architecture is illustrated in Appendix B.1. The flattened target image and the patient specific parameters were passed separately through one block each, concatenated, and passed through another block, after which the amplitude is predicted. The amplitudes were concatenated to the prior intermediate representation, fed through a final block, after which frequency and pulse duration were predicted. The output layers use ReLU activation; all others use leaky ReLU. During training, $\phi$ were randomly sampled from the range of allowed parameters (Table 1). Tensorflow 2.12, an NVIDIA RTX 3090, Adam optimizer, and batch size of 256 [38, 39] were used to train the network.

**Human-in-the-Loop Optimization** We propose using preferential Bayesian optimization (PBO) to optimize the patient-specific parameters $\phi$ of the pretrained DSE. Given two sets of patient-specific parameters, $\phi_1$ and $\phi_2$, we assume that the probability of a subject preferring $\phi_1$ to $\phi_2$ (returning a response $\phi_1 \succ \phi_2$) depends on a preference function $g(\phi)$, modeled using a Gaussian process model:

$$P(\phi_1 \succ \phi_2 | g) = \Phi\big(g(\phi_1) - g(\phi_2)\big), \tag{4}$$

where $\Phi$ is the normal cumulative distribution [40, 41]. The larger the value of $g(\phi_1)$ relative to $g(\phi_2)$, the higher the likelihood that the subject reports preferring $\phi_1$ over $\phi_2$.

We used the Maximally Uncertain Challenge [42] to select new comparisons to query, although other popular acquisitions performed similarly (Appendix C.3). Searching within the bounds in Table 1, this acquisition function selects a 'champion', $\phi_1$, which maximizes the expectation of $g$, and a 'challenger', $\phi_2$, for which subjects' preferences are most uncertain:

$$\phi_1 \to \arg\max_{\phi} \mathbb{E}_{p(g|\mathscr{D})}[g(\phi)], \tag{5}$$

$$\phi_2 \to \arg\max_{\phi} \mathbb{V}_{p(g|\mathscr{D})}[\Phi(g(\phi) - g(\phi_1))], \tag{6}$$

where $\mathbb{V}$ denotes the variance. This algorithm is designed to balance exploitation (values of $\phi$ that maximize $g$) and exploration (values of $\phi$ for which the response is uncertain).

The performance of PBO crucially depends on the Gaussian process kernel and its hyperparameters, which encode our prior assumptions about the latent preference function. Inferring the kernel's hyperparameters online would slow down the algorithm and could lead to overfitting. Thus, we adopted a transfer learning strategy, which could also be applied to real-life patients (see Appendices C.1 and C.2 for full details and discussion of alternatives). In brief, for each of 10 patients (with parameters different from those used in the following PBO experiment), we simulated 600 random duels and fit candidate hyperparameters for each of 4 commonly used kernels. We then selected the kernel and hyperparameters that generalized best to the other 9 patients (measured using Brier score [43] on a held-out test set). The 5/2 Matérn kernel performed best, and was used for all subsequent experiments.

**Simulated Patients** *In silico* experiments on simulated patients were used to demonstrate the viability of our approach. Each patient was assigned a set of patient-specific parameters $\phi$, uniformly sampled from the ranges specified in Table 1. When challenged with a duel between two candidate stimuli $\mathbf{s}_{\phi_1}$ and $\mathbf{s}_{\phi_2}$, the simulated patient ran each stimulus through the phosphene model (using ground-truth patient-specific parameters), obtaining the predicted percepts $\hat{\mathbf{t}}_{\phi_1} = f(\mathbf{s}_{\phi_1}; \phi)$ and $\hat{\mathbf{t}}_{\phi_2} = f(\mathbf{s}_{\phi_2}; \phi)$. The users' preferences were modeled with a Bernoulli distribution, with probability $p$ modulated by the difference in reconstruction error between each percept and the target image:

$$p = \frac{1}{1 + \exp(-\frac{1}{\sigma}(d(\hat{\mathbf{t}}_{\phi_2}, \mathbf{t}) - d(\hat{\mathbf{t}}_{\phi_1}, \mathbf{t})))} \tag{7}$$

Here, $\sigma$ is a configurable parameter that scales the width of the sigmoid, introducing noise into the response. We set $\sigma$ to be 0.01, chosen empirically based on a conservative estimate: when the error difference was greater than 0.01 it was obvious which percept was better to human observers.

**Data and Metrics** We used MNIST images as target visual percepts throughout the experiments. Images were resized to be the same size as the output of $f$ ($49 \times 49$ pixels), and scaled to have a maximum brightness of 2 (aligned with range($f$)). Inspired by [12, 14], we used a perceptual similarity metric designed to capture higher-level differences between images [44]. Let $v_l(\mathbf{t})$ be a function that extracts the downstream representations of target $\mathbf{t}$ input to a VGG19 network pretrained on ImageNet [45]. The perceptual metric is then given by equation 8.

$$d(\mathbf{t}, \hat{\mathbf{t}}) = \frac{1}{|t|}(||\mathbf{t} - \hat{\mathbf{t}}||_2^2 + \beta ||v_l(\mathbf{t}) - v_l(\hat{\mathbf{t}})||_2^2) \tag{8}$$

This metric was used by the deep stimulus encoder as a training objective, by the simulated patient to choose a duel winner, and throughout HILO as an evaluation metric. $\beta = 2.5\text{e-}5$ was selected via cross-validation (see [14] App. B). To aid in interpretability, we also report a secondary metric based on how identifiable the predicted percepts were. We first pretrained a separate deep net to 99% test accuracy on MNIST classification. We then measured the accuracy of this classifier on the predicted percepts at every iteration of HILO.

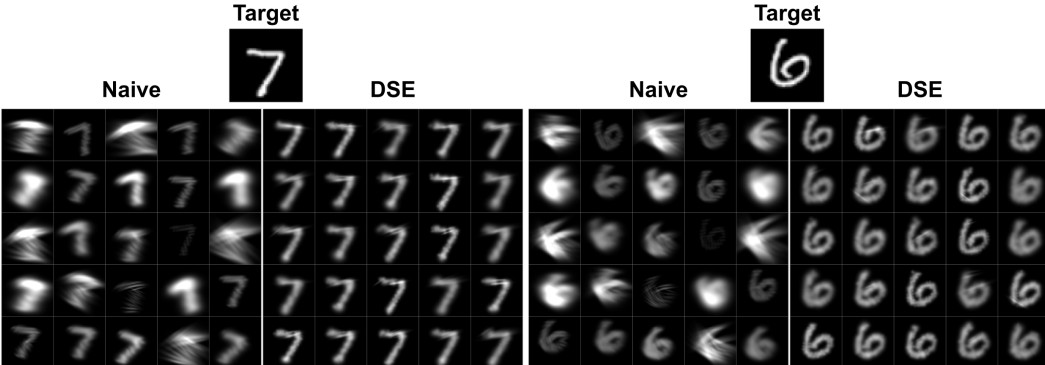

Figure 3: Percepts resulting from a naive encoder and the trained DSE for two example target images across 25 randomly selected patients.

## 4 Results

### 4.1 Phosphene Model

To verify that our phosphene model's predictions line up with observed results from real prosthesis users, we repeated analyses from previous state-of-the-art models, evaluating how phosphene appearance changes with electrode location [8] and stimulus parameters [10, 11, 19, 37]. We used the same datasets, consisting of thousands of phosphene drawings and brightness and size ratings collected across multiple epiretinal prosthesis [3] patients over several years. To evaluate phosphene appearances with electrode location, we calculated the correlation between predicted and observed phosphenes ($R_i$) for three shape descriptors: area, eccentricity, and orientation. The final score reported is $1 - \sum_i R_i^2$ [8]. To evaluate how phosphene appearance was modulated by stimulus parameters, we calculated the mean squared error between the size and brightness of predicted percepts and patient ratings as amplitude, frequency, and pulse duration were varied. The reported values correspond to Figures 4a–c and 5 in [11].

Evaluation results are presented in Table 2. Our model significantly outperforms all baselinse on the Beyeler *et al.* evaluation, and matches SOTA on the Granley *et al.* evaluations. Additionally, our model is on average 45x faster, and uses 120x less memory than [11]. A detailed description of evaluation methods and additional analysis can be found in Appendix A.2.

Table 2: Evaluation of Phosphene Model

| Model | Electrode Location [8] | | | Stimulus Parameters [11] | | | |
|---|---|---|---|---|---|---|---|
| | S1 | S2 | S3 | 4A | 4B | 4C | 5 |
| Nanduri *et al.* [19] | 37.3 | 19.5 | 294.8 | 11.1 | 72.9 | .2 | 160.9 |
| Beyeler *et al.* [8] | 2.43 | 7.07 | 1.15 | 215.6 | 108.7 | 5.52 | 190.2 |
| Granley *et al.* [11] | 2.43 | 7.07 | 1.15 | 0.9 | **2.1** | 0.16 | 49.5 |
| Proposed | **0.28** | **0.57** | **0.38** | **0.73** | 2.3 | **0.1** | **48.6** |

### 4.2 Deep Stimulus Encoder

We trained a deep stimulus encoder (DSE) to invert our phosphene model (decoder). The encoder was trained across 13 patient-specific parameters, randomly sampled at every epoch, including for the first time implant position and rotation. This is in contrast to previous DSEs, which either require retraining for every new patient [12, 13, 27], or can only vary two patient-specific parameters [14].

We compared the performance of the DSE to a traditional ('naive') encoder [14] currently used by retinal prostheses [3], illustrated in Fig. 3. The DSE achieved a test perceptual loss of 0.05 and a MNIST accuracy of 95.6%, significantly outperforming the naive encoder (5.68 and 51% respectively). Note that this performance is when the true patient-specific parameters are known.

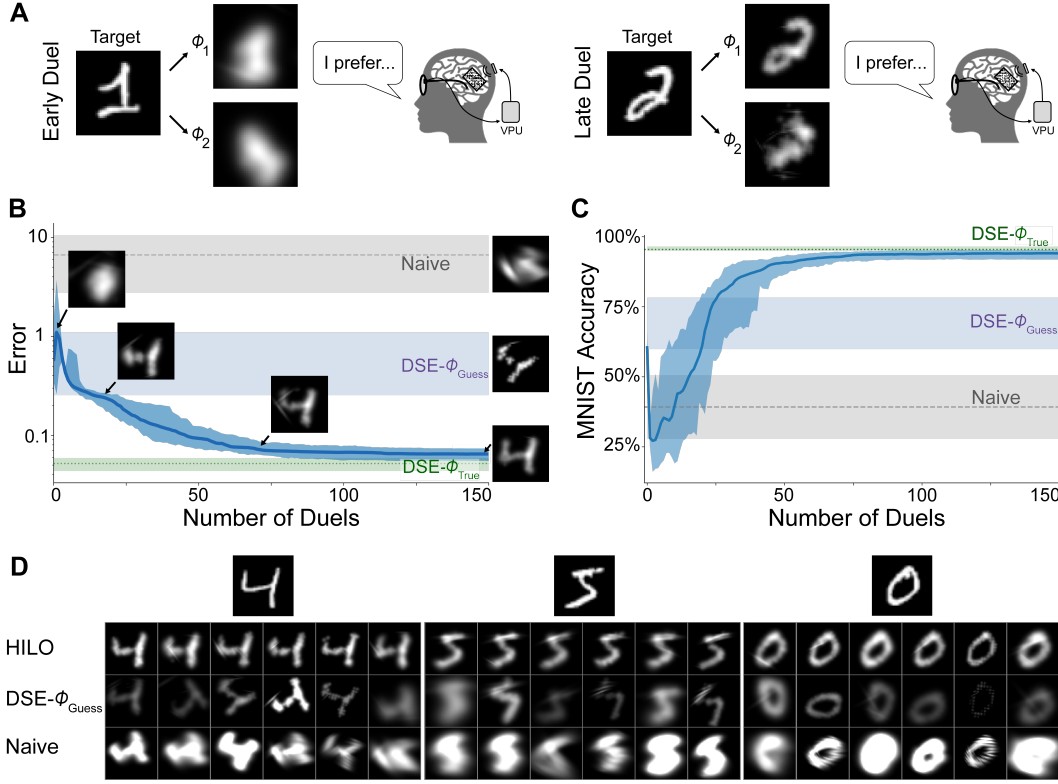

Figure 4: Human-in-the-loop optimization of a deep stimulus encoder. *A*: Two example duels, from which patient preferences are learned. *B*: Reconstruction error throughout optimization across 100 simulated patients. Insets show the predicted percept resulting from stimulation with various encoders. Note the y axis is on a log scale. *C*: MNIST accuracy of a pretrained classifier on reconstructed phosphenes. Both plots show smoothed median (window size of 3), with error bars denoting IQR. *D*: Example percepts after optimization for Naive, DSE without HILO, and HILO encoders for 6 random patients. Note, brightness is capped for display for the Naive encoder.

This performance was slightly better than the values reported in [14] (L1 loss of .108 vs .12 in [14]) despite training across 11 additional patient-specific parameters.

## 4.3 Human-in-the-Loop Optimization

We ran deep learning-based HILO for 100 randomly sampled simulated patients. After every duel, we evaluated the DSE parameterized by the current prediction of patient-specific parameters on a subset of the MNIST test set. The performance of the learned encoder over time ('HILO') is illustrated in Figure 4, which plots the joint perceptual loss (Figure 4.B) and MNIST accuracy (Fig. 4C).

As baselines for comparison we used a naive encoder, a non-personalized DSE where the patient-specific parameters are guessed (DSE-$\phi_{Guess}$), and an ideal DSE using the true $\phi$ (DSE-$\phi_{True}$). To guess $\phi$, we consider two approaches, one which selects the midpoint from the ranges in Table 1, and another where performance was averaged across random selections of $\phi$ from the same ranges. In real patients, we estimate that the performance of a deep stimulus encoder without patient-specific optimization would likely fall somewhere between these two methods, since the distribution of real patients might deviate slightly from Table 1. We therefore plot the region bounded by the performance of a DSE with either of these approaches for guessing $\phi$. Example percepts after optimization are shown in Fig. 4D.

The HILO encoder starts with random predictions, but, after a short initial exploration period, quickly surpassed the baselines. After about 75 iterations, performance approached the ideal DSE encoder, however the HILO encoder still resulted in high-quality percepts after as few as 20 iterations. Averaged across patients, the final reconstruction error of the HILO encoder was .071 ± .0031 (SEM)

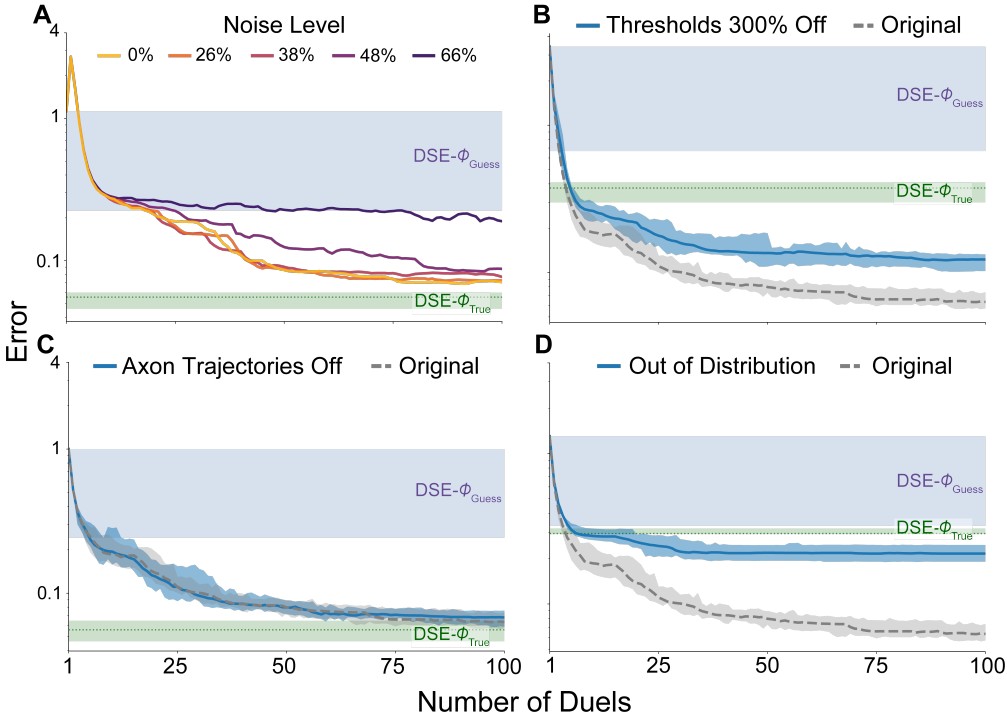

Figure 5: Reconstruction error through optimization for noisy patient responses (*upper left*) and for various misspecifications in the forward model assumed by the DSE. Noise level denotes the percentage of duels where the decision was essentially random ($p \in [0.35, 0.65]$), corresponding to $\sigma$ of 1e-4, 0.005, 0.01, 0.02, and 0.05, respectively. All y axes are on log scales. Naive encoders and some error bars omitted for clarity.

and MNIST accuracy was $92\% \pm 1.0\%$. DSE-$\phi_{Guess}$ had an error of between .25 and 1.1 and MNIST accuracy between 58.6% and 78.3%, and the DSE with true $\phi$ had an error of $.05 \pm .001$ and accuracy of $95.5\% \pm .1\%$.

### 4.4 Robustness

In reality, it is likely that a patient's perceptions will not be perfectly captured by the phosphene model. Further, patient responses for visual prostheses are notoriously noisy [7, 46]. To test HILO's resiliency to these variations, we conducted additional robustness experiments, each with the same 25 simulated patients (Figure 5). First, we varied the noise parameter $\sigma$ in simulated patients' decision making (Figure 5A). Next we constructed various 'misspecified' forward models, where the ground-truth model used to decode stimuli differed from the forward model assumed by the DSE. First, we varied the trajectories of the simulated axon bundles [47], thereby changing the orientation of phosphenes (Figure 5C). Second, threshold amplitudes for stimulation are notoriously hard to predict, and have been shown to drift by up to 300% over time [48]. Therefore, we tested a variant where the threshold assumed by the encoder was incorrect by up to 300% (Figure 5B). Lastly, we used the same forward model, but with patient-specific parameters outside the ranges in 1 (Figure 5D).

At $\sigma$=1e-4, the patient response was noiseless. For $\sigma$ equal to .005, .01, and .02, HILO performed similarly to the noiseless model, despite the patient on average making 'random' ($p \in [0.35, 0.65]$) decisions in 26%, 38%, and 48% of duels. At $\sigma$=0.05, the decision was 'random' 2/3 of the time, and HILO performed similarly or slightly better than the baseline DSE-$\phi_{Guess}$. The DSE itself is very resilient to misspecifications in axon trajectory, so HILO performs similarly for this misspecification to the original patients. When thresholds varied, HILO still outperformed the baselines, but converged to slightly worse encodings than without misspecification. Further, HILO surpassed the DSE encoded with the ground-truth $\phi$. This demonstrates that even when the forward model assumed by the DSE is incorrect, HILO still tends to converge to a set of patient-specific parameters that work well for the

misspecified patient. For out-of-distribution $\phi$, HILO again outperformed both the baseline and true DSEs, but performed worse than in-distribution patients.

## 5 Discussion

Our experiments show that HILO optimization of a deep stimulus encoder led to high-quality, personalized stimulation strategies that outperformed previous state-of-the-art techniques. HILO led to an increase in percept quality compared to using a non-personalized DSE for 99 out of 100 simulated patients, demonstrating the viability of our approach. To enable our HILO algorithm, we also developed a new phosphene model, which is computationally simpler and matches patient data better than previous models, and trained a new DSE, which is able to produce high-quality encodings across all 13 dimensions of patient-specific variations included in our phosphene model. Together, these significantly advance state-of-the-art in patient-specific stimulus encoding, and are important steps towards practically-feasible personalized prosthetic vision in real patients.

The proposed framework combining Bayesian optimization and deep stimulus encoding offers significant improvements over both components in isolation. Use of a DSE allows us to incorporate prior information, reducing the dimensionality of the Bayesian optimization search space from the large stimulus space to the much smaller model parameter space. Our results demonstrate that even when the DSE's predictions are incorrect, this parameterization is still useful for Bayesian optimization based on patient preferences. Additionally, DSEs are able to invert highly nonlinear forward models, enabling encoder-parameterized Bayesian optimization to be applied to a much larger set of problems. Lastly, the learned encoder can be applied for any target percept, without needing additional optimization. On the other hand, without adaptive feedback from HILO, deep stimulus encoders have no method for learning the individual differences of a new patient, which we show leads to suboptimal stimuli. DSEs rely on the accuracy of their assumed forward model over the entire stimulus space. We show that our approach produces stimuli that work well for the patient, even when the forward model is misspecified, or when the patient's responses are noisy.

This approach is practical for stimulus optimization in the wild. The encoder learned during optimization is lightweight, and once deployed, can predict individual stimuli in less than $5\,\mathrm{ms}$ on CPU, allowing for high frame rates for prosthetic stimulation. During HILO, updating the Gaussian process model and producing new stimuli on average took 3 seconds, meaning that stimulus optimization could be performed in a matter of minutes. A HILO strategy could be bundled with future visual prostheses, allowing for patients to periodically re-calibrate their devices when they feel the device is not performing adequately, without requiring a clinical professional.

**Broader Impacts** Although we demonstrate this approach in the context of visual prostheses, our framework is general and could be applied to a variety of sensory devices. Our approach is applicable when the stimulus search space is large and there exists a forward model mapping stimuli to responses. Forward models [49–52] and deep stimulus encoders [53–55] have been successfully used across multiple sensory modalities, and could potentially be adapted for personalization with HILO.

**Limitations** Although promising, our approach is not without limitations. We assumed that the preference of patients for different stimuli is related to the distance metric used to measure perceptual similarity, which may not be true in practice. However, results by [35] suggest that PBO is robust to a mismatch between the distance metric used to invert the forward model and the preference of patients. Another limitation is that evaluation of our approach was only performed on simulated patients with a simulated perceptual model. However, this is mitigated by the fact that HILO showed robustness to model inaccuracies. Still, since it is difficult to predict the behavior of deep learning models, using a deep stimulus encoder in real patients could raise safety concerns. It may be possible for a deep encoder to produce unconventional stimuli, potentially leading to adverse effects. However, most devices come with firmware responsible for ensuring stimuli stay within FDA-approved safety limits.

In conclusion, our results suggest that combining the strengths of deep learning and Bayesian optimization could significantly improve the perceptual experience of patients fitted with visual prostheses and may prove a viable solution for a range of neuroprosthetic technologies.

# 6    Acknowledgements

Research reported in this publication was supported by the National Library of Medicine of the National Institutes of Health under Award Number DP2LM014268. The content is solely the responsibility of the authors and does not necessarily represent the official views of the National Institutes of Health.

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

# Appendix

## A  Phosphene Model

### A.1  Methods

This section describes the phosphene model used to simulate patient's perception resulting from stimulation. The model takes in a stimulus vector $s \in \mathbb{R}^{n_e \times 3}$ specifying the frequency ($freq$), amplitude ($amp$), and pulse duration ($pdur$) of a biphasic pulse train on each electrode. In addition, the model also takes in a vector of patient-specific parameters $\phi$ (see Table 1). We break these parameters down into implant parameters ($x, y, rot$), global parameters ($\rho, \lambda, \omega, OD_x, OD_y$), and stimulus-related parameters ($a_0$-$a_4$); all explained below.

Exact implant locations vary patient-to-patient. The three implant parameters allow our model to account for these changes. We used a simulated implant inspired by designs of real epiretinal implants [3, 56] and those used in previous simulation studies [14]. It consists of 225 disk electrodes (radius $75\,\mu$m arranged onto a square, $15 \times 15$ grid with $400\,\mu$m spacing, initially centered over the fovea. The three implant-related parameters translate and rotate the initial implant to be centered at $(x, y)$, and to be rotated by $rot$ degrees. The implant used is depicted in Figure A.2, overlaid on top of a simulated map of axon nerve fiber bundles [57].

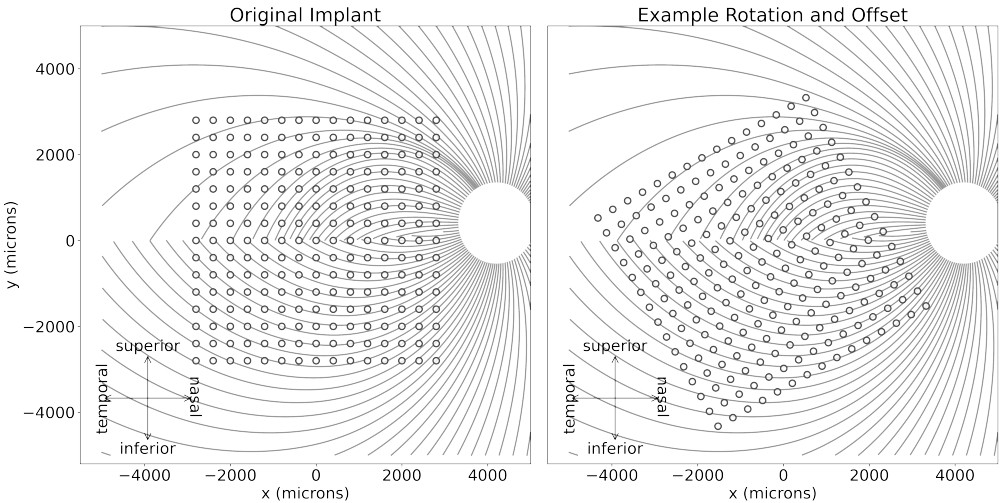

Figure A.1: The implant used for optimization, and an example implant after rotation and translation based on patient-specific parameters $\phi$. The white circle on the right is the optic disc. Arced lines depict simulated axon nerve fiber bundles.

The remaining model parameters are inspired by various psychophysical and electrophysiological studies [8, 10, 18, 19, 58], and are summarized in the following list:

- $\rho$ : Average phosphene size. This will be modified locally based on stimulus parameters.
- $\lambda$ : Average phosphene eccentricity (a measure of phosphene elongation; not to be confused with retinal eccentricity). This will be modified locally based on stimulus parameters.
- $\omega$ : Orientation scaling factor. The orientation of phosphenes will be the orientation of the underlying axon bundle, scaled by $\omega$ (Eq. 12).
- $OD_x, OD_y$: The x and y location of the patient's optic disc, into which axon nerve fiber bundles terminate.
- $a_0$-$a_2$: Coefficients to modulate phosphene brightness with stimulus parameters (Eq. 9).
- $a_3$ : Coefficient to modulate phosphene size with stimulus parameters (Eq. 10).
- $a_4$ : Coefficient to modulate phosphene eccentricity with stimulus parameters (Eq. 11).

Each electrode's location on the retina can be determined from the implant parameters. The corresponding location in the visual field ($\mu_e$) is determined using the retinotopic map described in

Watson *et al.* [59]. Each electrode's phosphene orientation is then $\theta_e = \omega\theta_{axon}$, where $\theta_{axon}$ is the orientation of the axon nerve fiber bundle (NFB) underlying the cell (pixel). Axon NFBs are modeled as spirals originating at the optic disc and terminating at each simulated cell. These spirals follow a simulated axon map [47] based on tracings of axon trajectories in 55 human eyes. In summary, phosphene size, eccentricity, brightness, and orientation are modulated based on stimulus parameters and implant location according to the following equations:

$$b_e = a_0(amp_e)^{a_1} + a_2(freq_e) \tag{9}$$

$$\rho_e = \rho * a_3 * amp_e \tag{10}$$

$$\lambda_e = \lambda \left(\frac{pdur}{0.45}\right)^{a_4} \tag{11}$$

$$\theta_e = \omega * \theta_{axon} \tag{12}$$

The phosphene for each electrode is a multivariate Gaussian blob, centered at the electrodes location in visual field ($\mu_e$), and with covariance matrix $\Sigma_e$ constructed such that the resulting phosphene will have brightness $b_e$, size $\rho_e$, eccentricity $\lambda_e$, and orientation $\theta_e$, as shown in the following equations (repeated from main text for convenience):

$$b(x, y) = 2\pi b_e \det(\mathbf{\Sigma_e}) \, \mathcal{N}([x, y]^\top | \mu_e, \mathbf{\Sigma_e}), \tag{13}$$

The covariance matrix $\mathbf{\Sigma_e} = \mathbf{R}\mathbf{\Sigma_0}\mathbf{R}^T$ is calculated from the eigenvalue matrix $\mathbf{\Sigma_0}$ and a rotation matrix $\mathbf{R}$:

$$\Sigma_0 = \begin{bmatrix} \alpha_e^2 & 0 \\ 0 & \beta_e^2 \end{bmatrix}, \qquad R = \begin{bmatrix} \cos\theta_e & -\sin\theta_e \\ \sin\theta_e & \cos\theta_e \end{bmatrix}.$$

The eigenvalues $\alpha_e$ and $\beta_e$ depend on the intended phosphene area ($\rho_e$) elongation ($\lambda_e$), and a constant $\epsilon$ (set to $e^{-2}$):

$$\alpha_e^2 = -\frac{\rho_e\sqrt{1 - \lambda_e^2}}{2\pi \ln \epsilon}, \qquad \beta_e^2 = -\frac{\rho_e}{2\pi \ln \epsilon \sqrt{1 - \lambda_e^2}}.$$

This formulation guarantees that the Gaussian blob, when thresholded using $\epsilon$, will have the intended area, orientation, eccentricity, and brightness.

Blobs from individual electrodes are summed into a global percept. This linear summation is supported by recent studies, which have shown that percepts from multi-electrode stimulation are often linearly related to the percepts from stimulation on the individual electrodes [36]. Although the sum across electrodes is linear, modulating the size and eccentricity of phosphenes makes the final result a nonlinear function of stimulus parameters, preventing analytic inversion.

## A.2 Evaluation

Our model is motivated by similar anatomical and psychophysical phenomena as the previous state-of-the-art model for epiretinal prostheses [11], but its formulation allows for favorable computational properties. In comparison, our model is on average 45x faster to run, and consumes about 120x less GPU memory. These computational benefits are the main reason a new model was necessary, and enables a more advanced deep stimulus encoder by allowing training with larger encoder models, longer training duration, and larger batch sizes.

Nonetheless, we also verified that the model produces state-of-the-art predictions, as described in Section 4.1. Despite its similar design, our model achieves much better scores on the Beyeler *et al.* [8] evaluation for phosphene shape. This is likely because our formulation allows much tighter control of phosphene shape attributes (e.g., size, eccentricity), allowing (for the first time) positive $R^2$ on shape descriptors for held-out electrodes. Our model performs similarly to the previous state-of-the-art model on the Granley *et al.* [11] evaluation, which is to be expected given that the equations modulating phosphene appearance with stimulus parameters in both models are very similar.

Fig. A.2 reproduces the plots from Figures 4A-C and 5 of [11], but with our proposed model included. These figures show the brightness or size rating from Argus II patient(s) as stimulus parameters vary [19, 37], in subjective units. For brightness, '10' means the same as the reference pulse, '20' means twice as bright, etc. For size, '1' means the same as reference pulse, '2' means twice as large (notation matching [11, 19, 37]).

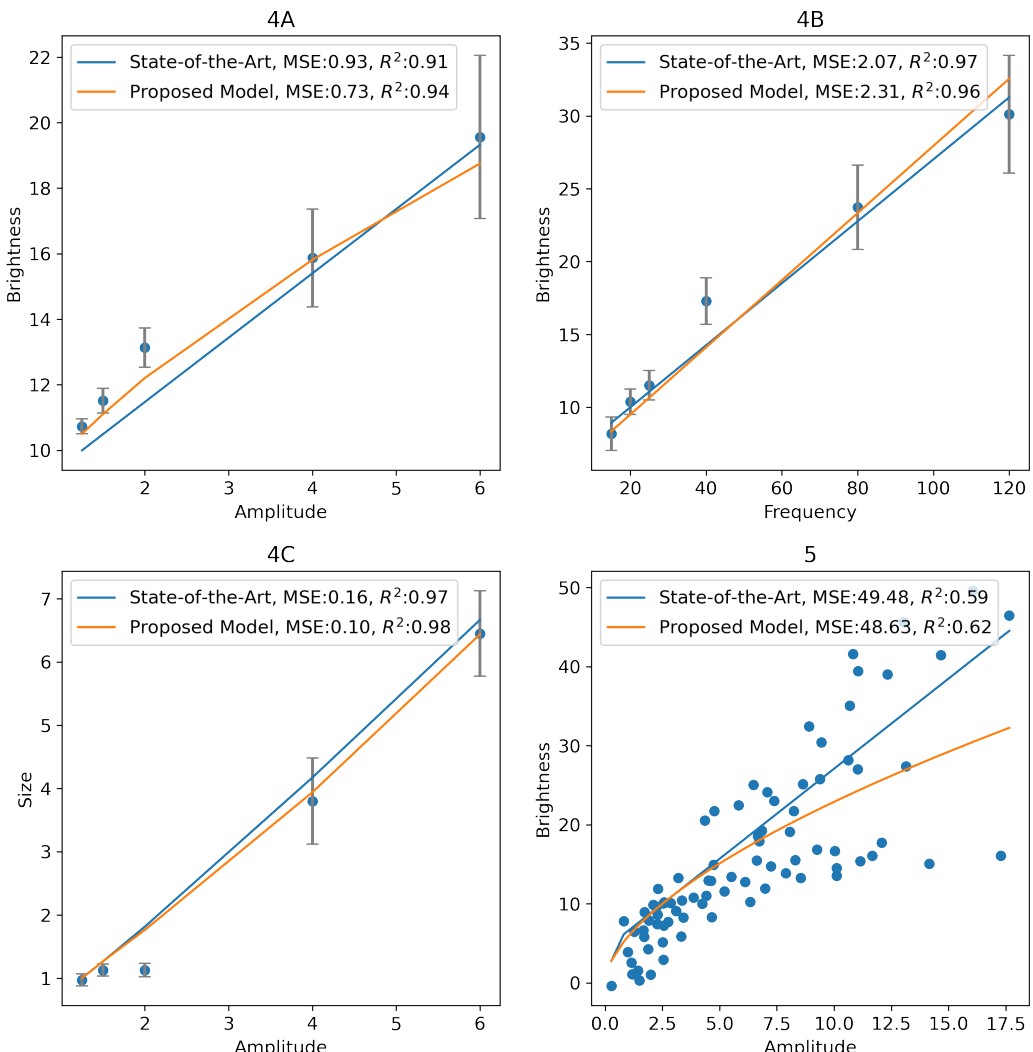

Figure A.2: Evaluation of phosphene brightness and size as stimulus parameters vary. Reproduced from [11], but with our proposed phosphene model included. State-of-the-art denotes the phosphene model from [11]. Units are subjective, in comparison to a reference pulse (i.e. brightness of 20 means twice as bright, size of 3 means 3 times as bright) [19]

# B    Deep Stimulus Inversion

## B.1    Architecture

The encoder architecture described in Section 3 is illustrated in Fig. B.1.

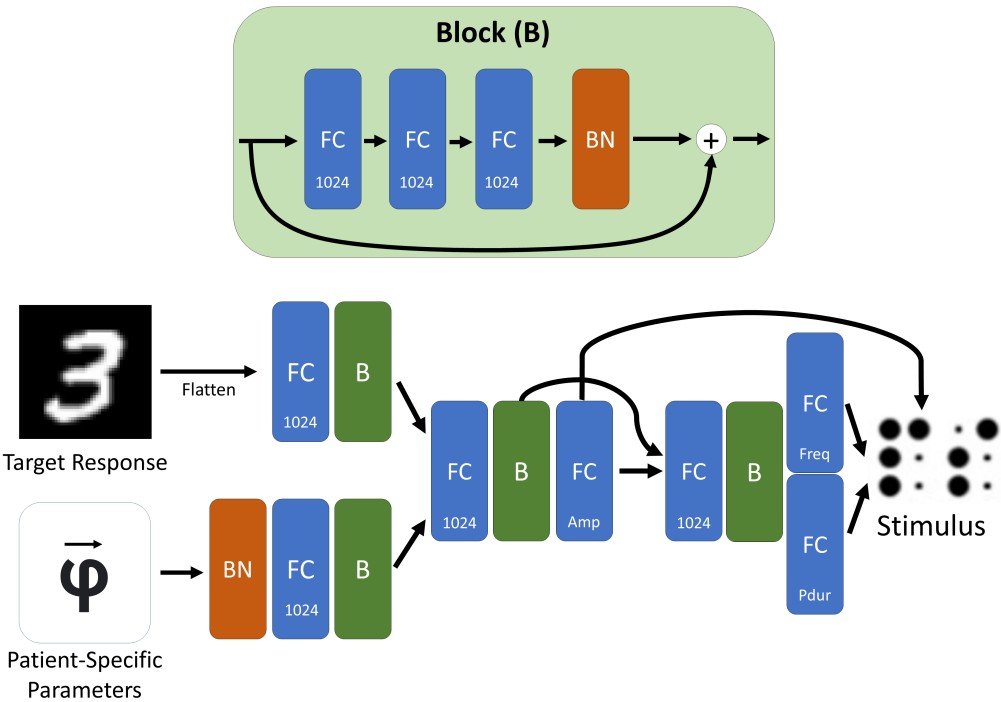

Figure B.1: Deep stimulus encoder architecture. FC: fully connected layer, BN: batch normalization layer, B: block of layers (shown at top). Two arrows merging into one fully connected layer denotes concatenation. 45M total parameters.

# C   Human-in-the-Loop Optimization

## C.1   Kernel Selection and Hyperparameters

This section gives more details on fitting the hyperparameters for the Gaussian process (GP) kernel used in preferential Bayesian optimization (PBO). As stated previously, the performance of PBO crucially depends on the GP kernel and its hyperparameters, which encode our prior assumptions about the latent preference function. To select hyperparameters, we used a transfer learning strategy, selecting hyperparameters that generalized best within a small validation group of simulated patients (see Appendix C.2 for discussion).

We simulated 600 random duels ($\phi_1$ and $\phi_2$ chosen randomly) on each of 10 simulated patients. For each patient, we fit four commonly used kernels (Squared Exponential, Squared Exponential with Automatic Relevance Determination (ARD), Matérn 3/2, and Matérn 5/2) and inferred hyperparameters using type II maximum likelihood estimation [60]. The bounds for each hyperparameter were $[\exp(-10), \exp(10)]$. For each of these candidate kernel-hyperparameter pairs, we fit a GP with the corresponding kernel and hyperparameters to 50 training duels for each of the other 9 patients. Then, the performance of the candidate GP was evaluated on the remaining 550 data points using Brier score [43], a commonly used metric measuring the accuracy of probabilistic predictions:

$$BS = \frac{1}{n}\sum_{i=1}^{n}(y_{true} - y_{pred})^2,\tag{14}$$

where $y_{true}$ is the true duel outcome (1 or 0) as decided by the simulated patient, and $y_{pred}$ is the probability of $\phi_1$ being selected as the winner (corresponding to outcome of 1), as predicted by the Gaussian process.

The kernel and hyperparameters with the lowest Brier score, averaged across all 9 other patients, were selected (Matérn 5/2). To verify that this kernel performed well, we also ran human-in-the-loop optimization (HILO) for 20 random simulated patients, using the best hyperparameters for each of the four previously mentioned kernels. The results are shown in Figure C.1. The Matérn 5/2 kernel performed slightly better than the Matérn 3/2 kernel, and significantly better than the ARD kernel. While performance was similar to the Squared Exponential kernel, we ultimately selected Matérn 5/2 due to its lower Brier score.

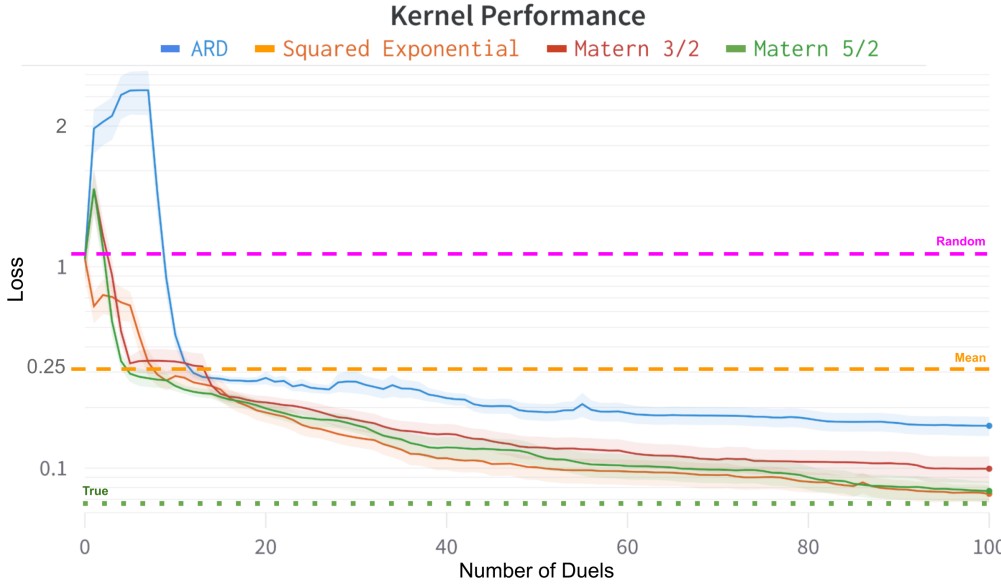

Figure C.1: Joint perceptual loss (y axis, log scale) plotted throughout HILO with different Gaussian process kernels. Error bars denote SEM.

## C.2 Hyperparameter Optimality

The Gaussian process kernel hyperparameters selected with our transfer learning strategy performed well in our simulations, leading to higher-quality patient-specific stimulus encodings (Figure 4). This transfer learning approach was chosen to match a clinical setting, where limited human data availability. However, it is certainly possible that better performing hyperparameters exist. To investigate the optimality of our hyperparameters, we examine two other strategies: an ideal case where 'patient-optimal' hyperparameters are used for each patient, and an online strategy where hyperparameters are updated during optimization with each patient,

**Patient-optimal Hyperparameter Selection**    In this strategy, hyperparameters were precomputed for each patient by simulating 200 random duels. Kernel parameters were again fit using type II maximum likelihood estimation [60]. These 'patient-optimal' hyperparameters were then used during HILO for the patient.

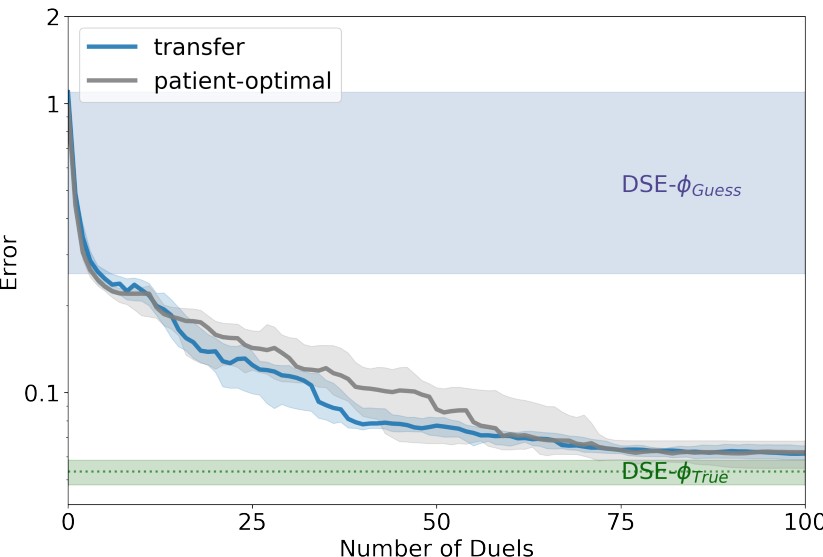

Figure C.2: Reconstruction error using Gaussian process kernel hyperparameters selected via transfer learning and 'patient-optimal' strategies throughout HILO.

Optimization results for 20 random patients are shown in Figure C.2. Both the transfer and the patient-optimal settings led to similar performance, and there was no significant difference between the final reconstruction errors ($p > .05$, two-sided paired t-test). Thus it seems the transfer learning strategy indeed selected hyperparameters that generalize well for new patients. Note that in general the transfer learning selection strategy is more practically applicable than the patient-optimal strategy since it does not require a long calibration period, but the patient-optimal strategy is an alternative that could be used for the first human subjects when no human data (only simulated data) is available.

**Online hyperparameters selection**    While it is common to keep kernel hyperparameters constant during optimization [33, 61, 62], online optimization is an alternative, where kernel parameters are periodically re-fit to the patient data during optimization. We tested online optimization where hyperparameters were recalculated with an update period of 1, 5, 10, or 20 duels, or were never updated. In all settings, the initial hyperparameters were chosen using the transfer learning strategy.

The results for online hyperparameters optimization are shown in Figure C.3. All update periods eventually converged to similar performance as with the transfer learning strategy (*i.e* never updating hyperparameters), but since recalculating hyperparameters is costly, online hyperparameter updates increased the optimization time required to reach a desired performance (Figure C.3, *right*). This suggests that just using the transfer learning hyperparameters is a better strategy than online updates.

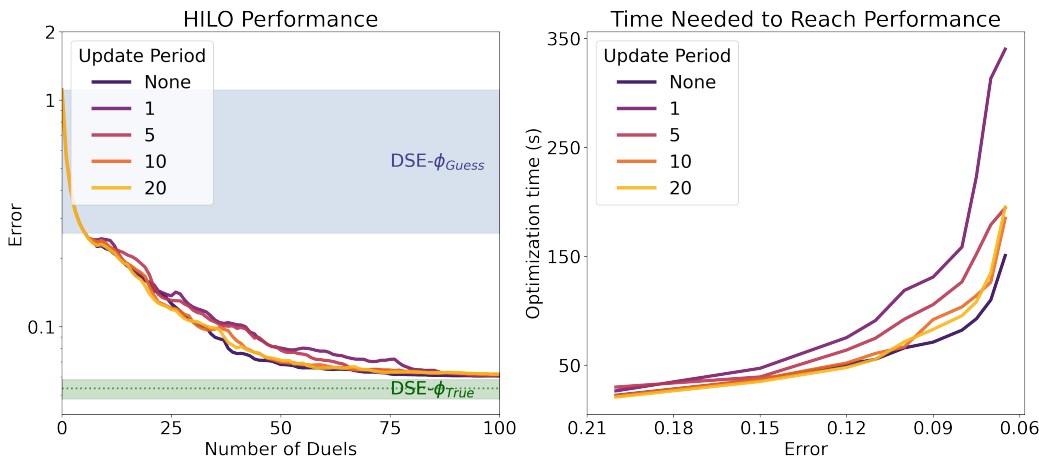

Figure C.3: Reconstruction error (left) and Bayesian optimization time required to reach a specified reconstruction error (right) for HILO with online updates of GP hyperparameters.

## C.3 Acquisition function

The acquisition function is responsible for choosing $\phi_1$ and $\phi_2$ for each duel, and must balance exploration of the search space with exploiting values of $\phi$ that are expected to work well. The Maximally Uncertain Challenge (MUC) [42] acquisition presented in the main text was initially compared against 2 other top-ranking [42] acquisition functions: Bivariate Expected Improvement (Bivariate EI) [63], and Dueling Upper Credibility Bound (Dueling UCB) [64]. In addition, we also compare against a baseline acquisition, where $\phi_1$ and $\phi_2$ are chosen randomly for each duel. We ran HILO for the same 20 random patients for each acquisition.

The joint perceptual loss throughout optimization for each acquisition is presented in Figure C.4. All of the tested acquisition functions dramatically outperformed the random baseline, which converged to a value near the mean DSE without HILO. Although MUC and Dueling UCB performed similarly, we ultimately selected MUC due to its slightly lower final loss.

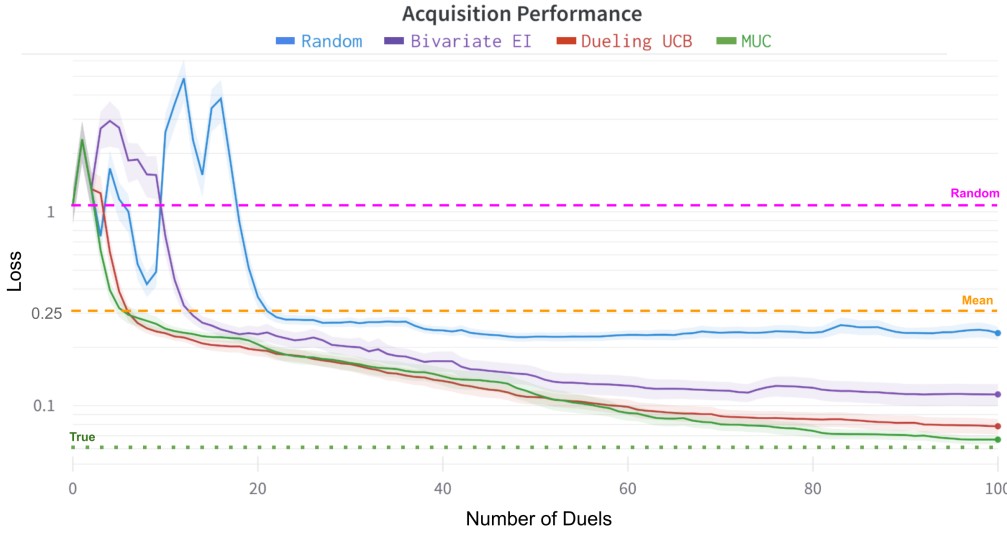

Figure C.4: Joint perceptual loss (y axis, log scale) plotted throughout HILO with different acquisition functions. Error bars denote SEM.

## C.4 Batch Bayesian Optimization

While the proposed Bayesian optimization strategy of sequentially presenting the user with stimuli would likely not be prohibitively time consuming (about 17 minutes with 100 duels, 10 seconds per duel), it is possible that batch Bayesian optimization could speed up optimization. We consider two formulations of batch Bayesian optimizations: 1) the user selects their preference from a batch of stimuli in each iteration, and 2) each duel still has only two options, but batches of duels are precomputed to save on updated the Gaussian process posterior and the acquisition function time between duels.

Option 1 is theoretically more ideal, since more information acquired in each comparison would hopefully allow for fewer comparisons. However, this would require the patient to remember an entire batch of stimuli before making a comparison, and in practice phosphenes can be very difficult to discriminate [7]. It is difficult to evaluate the effect of this cognitive burden on simulated patients, and we thus leave it to future work to consider whether the parallelization of data acquisition would make up for the increased difficulty of the task.

Therefore, option 2 was tested by precomputing batches of 1 (*i.e. original*), 3, 6, or 10 duels at a time. We used a batch variant of the maximally-uncertain challenge acquisition function [42]. Results for 20 simulated patients are shown in Figure C.5. Precomputing batches reduced the optimization time required to reach a desired performance, but at the cost of requiring more duels. In practice, the optimal batch size could be determined by balancing the time required per duel with the time required for optimization. Faster acquisitions (*e.g.* KernelSelfSparring [65]) could further reduce optimization time.

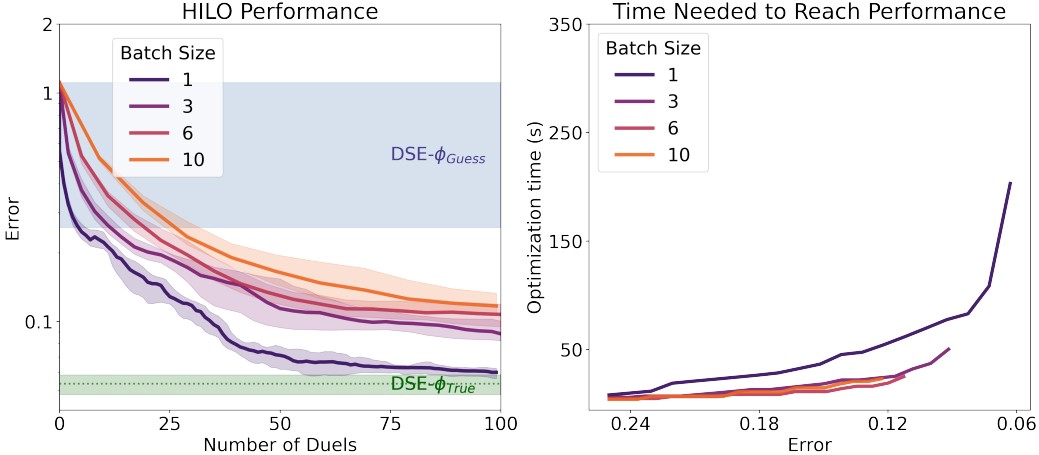

Figure C.5: Reconstruction error (left) and Bayesian optimization time required to reach a specified reconstruction error (right) for HILO with different batch sizes.

## C.5 Baselines and Robustness

In this section, we provide further details for the naive encoder we compared against, and the robustness experiments.

**Naive Encoder** The naive encoder is the encoding strategy currently used in commercial epiretinal prostheses [3]. This encoder operates under the assumption that each electrode can be thought of as a pixel in an image. The optimal stimulus under this assumption is therefore simply a downsampled version of the target image. The frequency and pulse duration are constant across all electrodes. This naive encoder has been previously shown to be suboptimal [14], but we still include it as a comparison to the currently used encoding strategy.

**Robustness** In section 4.4 we evaluate the robustness of human-in-the-loop optimization (HILO) to misspecifications in the forward model. Here, we provide specific details on the implementation of these misspecifications, and how we adapted the baseline encoders to the misspecified scenarios.

- **Axon Trajectories**: The simulated axon map from [47] has two parameters, $\beta_{inf}$ and $\beta_{sup}$, which control the axon trajectories in the inferior and superior retina, respectively. [47] also reports the observed ranges for these parameters: $\beta_{sup} \in [-2.5, -1.3]$ and $\beta_{inf} \in [0.1, 1.3]$. The unmodified model uses the centers of these ranges. Under misspecification, we randomly set both $\beta_{sup}$ and $\beta_{inf}$ to one of these bounds for each patient.

- **Thresholds**: Threshold is the amplitude at which a phosphene becomes visible to a patient $50\,\%$ of the time. In epiretinal prostheses, thresholds are notoriously noisy, and vary significantly across electrodes, patients, and over time [48]. While some progress has been made towards predicting these thresholds [66], most state-of-the-art models assume that thresholds are known.

    With this misspecification, the assumed threshold on each electrode was modified by a random but systematic amount. Specifically, the threshold on each electrode was randomly selected to be between $\frac{1}{2}$x and 2x its original value for the 100% condition and between $\frac{1}{4}$x and 4x its original value for the 300% condition.

- **Out of Distribution**: It is also possible that a new patient does not fall within our assumed ranges. Thus, we tested a variant where the true patient-specific parameters $\phi$ were sampled from outside the ranges in Table 1. Specifically, each parameter was sampled to be 0-50% above or below the specified range (some parameters were clipped to stay within defined ranges, *e.g.*, $\lambda$ cannot be outside of $[0, 1)$).

    During HILO, the acquisition functions have specified bounds that constrain candidate $\phi$. We therefore tested two variants, one where PBO was allowed to expand the bounds, and another where it was confined to within its original bounds. The end results were similar in terms of DSE performance, so the variant with its original bounds is presented in the main text.

DSE-$\phi_{Guess}$ is our best approximation of what a DSE would guessed patient-specific parameters would perform, and is bounded by the performance of a DSE with mean $\phi$, and random $\phi$ from the ranges in Table 1. For each of the misspecifications, the mean and random $\phi$ baseline DSEs are still encoded with the same $\phi$ as in the unchanged patient, but since the phosphene model for the patient is changed, the resulting loss is different. DSE-$\phi_{True}$ is still parameterized with the patients true $\phi$, however, the true $\phi$ are no longer a perfect description of the misspecified patient. This is shown by the fact that HILO surpasses the true encoders performance: under the misspecified model, there exists some other $\phi$ which leads to percepts with improved perceptual quality when decoded using the misspecified phosphene model, compared to those encoded with the true $\phi$. This highlights the robustness of optimization based on user preferences.

