# OpenReview forum: "Human-in-the-Loop Optimization for Deep Stimulus Encoding in Visual Prostheses"
_NeurIPS.cc/2023/Conference — NeurIPS 2023 poster_

### Official Review · Reviewer_HJZJ · 2023-07-04

**Soundness:** 3 good
**Presentation:** 4 excellent
**Contribution:** 2 fair
**Rating:** 6
**Confidence:** 4

**Summary:**

The authors present a method to improve perceptions for patients using visual prostheses using a combination of a deep learning encoding model and a patient-specific tuned set of parameters for stimulation learned via preferential Bayesian optimization. They use simulated data and patient choices to demonstrate the method’s performance across a range of possible patient types and noise considerations.

ETA: After review and rebuttal, raising my score from a 5 to a 6.

**Strengths:**

Originality: This work presents a nice combination of deep learning and Bayesian optimization. Each component of their approach (phosphene model, encoding model, and parameter optimization) has in general been done before, but here the authors modify each element and combine them to overall improve performance. The phosphene model is described as a new model with performance exceeding current SOTA. This was then combined with a deep stimulus encoder (DSE) to yield performance comparable to other methods. Finally, their human-in-the-loop (HILO) method to tune parameters using PBO was used to iteratively improve the DSE per patient.

Quality: In general the work is well-sourced and supported from many previous works. The benchmarking and evaluation of performance was good, and the authors compared to similar methods. The code in particular was very well written and documented.

Clarity: Overall well-written, well-sourced, nicely explained work. There were some points of confusion or ambiguity mentioned below.

Significance: This kind of patient-specific tuning of stimulation for visual prostheses is certainly needed, and the authors do good job of demonstrating the significance of their contribution. The authors also suggest it could be used in other areas beyond visual stimulation, but with the results being tied closely to the patient-specific parameters in Table 1 it is unclear how broadly this method might be applied.

**Weaknesses:**

The primary weakness is the reliance on simulated data, and it is not clear that simulated patient choice or patient parameters to generate the data was sufficiently grounded in experimentally collected datasets. There is some discussion of how the ranges for the parameters were chosen in their simulated data, but nothing about the expected variability and how these parameters were determined in the first place (is it considered ground truth? were patient preferences included?). If the method is sufficiently generalizable to many populations of patients, as expected, then why not simulated patients outside of those limited ranges in Table 1? Lines 263-265 appear to suggest that their ranges of parameters are unlikely to be same as in the real world. A similar concern is for simulating patient preference, but here the authors' simulations (up to 33% incorrect decisions from the patient) appear like a robust demonstration.

One other weakness not discussed is the feasibility of their approach for actual human patients. Even if it only takes 3 seconds to update the GP and generate new stimuli, the patient must sit through and compare difficult to discern visual percepts for 100 duels. Is it likely the patient would persist? Does their judgement change over time? If this could be changed every so often, does the system need re-tuning for all ~100 duels at once? Additionally, humans see visual scenes with certain image statistics -- so MNIST may not be a reasonable or sufficient dataset for testing human perception on. Could this approach be extended to natural images, including color?




**Questions:**

i.	Line 43 says <30 is too few parameters, but then the authors used only 13 dimensions of parameters?
ii.	Line 129-130 appears to have a sentence cut off abruptly.
iii.	Line 127: BO can’t optimize more than a small number of parameters -- clarify please? BO gets difficult in high dimensions, but this is true of many methods. How specifically does the DSE reduce this?
iv.	Clarify the issue in lines 154-158. The logic is not clear what the exact problem is and how it is solved here.
v.	If you didn’t restrict the range of parameters in Table 1 as heavily, would it still find the optimal values? E.g. if you increased the range by a factor of 2.
vi.	Line 232 first mention of Argus II, needs explanation and citation.
vii.	What does it mean to outperform the true DSE? Line 289
viii.	Line 296 how does the approach fail?

**Limitations:**

The authors addressed some limitations in the discussion, noting some drawbacks of predicting the behavior of deep learning algorithms but that hardware-engineered safety constraints would provide patient protection.

---

> ### Author Rebuttal · Authors · 2023-08-10
>
> We thank the reviewer for their positive review and insightful analysis.
> ## Weaknesses
> > Simulated Data and Patient Ranges
>
> We agree that proper selection of patient parameters is crucial for realism. The process of choosing these ranges of patient parameters was complicated by the variety of data sources [8,10,11,18,19,37,62], each using very different experimental setups.
> We used fits from the mentioned sources, based on psychophysical tasks such as drawings and brightness ratings, to obtain a distribution of possible values in patients. Our ranges were chosen empirically to encompass all observed patients with a wide additional margin (~2x the observed distribution), adjusted to be within allowed boundaries (e.g. $\lambda$ cannot be larger than 1).
> Therefore, the ranges in Table 1 are actually quite wide, more than account for all reported data, and only appear small due to the units. For example, $a2$ goes from 0.005 to 0.025, meaning that a frequency increase from 20 to 120 Hz would lead to a percept being between 1.5x and 3.5x as bright, a larger range than observed in patients [19].
> Unfortunately, we do not have time to retrain our DSE and run experiments again with increased ranges. However, HILO still led to large improvements for out of distribution patients (even without retraining, Fig 4), so we do not expect patients outside our ranges to prevent HILO from performing well.
> > 263-265 suggest that their ranges of parameters are unlikely to be same as in the real world.
>
> We only included this to acknowledge that any estimate is likely slightly different from reality. The only result that depends heavily on these ranges is the mean DSE baseline, which uses the mean value of the reported ranges as the guess for $\phi$. This is of course expected to work better on our simulated patients (from the exact same range) than on real patients (likely a similar, yet slightly different range).
> > Practical feasibility, changing perception, and natural images:
>
> The reviewer raises interesting questions. Based on the difficulty of standard clinical tasks [63] often involving hundreds of trials over multiple hours, we do not expect HILO to be too difficult. We also note that our method showed clear performance improvements after only 20 duels. Prior work suggests that phosphenes remain relatively stable over time [46]. If perception changes, HILO is short enough (~17 minutes if all 100 duels are needed, 10 seconds per duel) that it could easily be performed again.
> The reviewer is certainly correct that perceiving MNIST digits is not the eventual goal of visual prostheses. However, seeing even simple characters without prolonged head scanning would be a significant improvement in perception for existing devices [63]. DSE’s have already shown improvements for natural images [14,12], which could enable a similar approach to ours with complex targets. Unfortunately, detailed color perception is not possible with current visual prostheses [47].
>
> ## Questions
> > Line 43 says <30 is too few parameters...
>
> This is saying that PBO conventionally requires less than 30 parameters.
> > Line 127: BO can’t optimize more than a small number of parameters -- clarify please? … How specifically does the DSE reduce this?
>
> Although many optimization methods become difficult in higher dimensions, PBO quickly becomes inefficient with more than ~30 parameters [17]. In contrast, DNNs excel at high dimensional optimization. Our stimuli are high dimensional: 225 electrodes, each with 3 stimulation parameters. Further, we want to optimize stimuli not just for one target image, but for any target image in the training set. This is a regime where direct BO is clearly not possible.
> Our DSE is first trained to output optimal stimuli for any input patient (specified by $\phi$). Then, BO is used to infer $\phi$ for a new patient. Once known, the DSE can produce optimal stimuli for that patient. $\phi$ is low dimensional (13 parameters), and thus is a suitable candidate for BO, whereas the original stimulation pattern (675 parameters) is not. Thus, the DSE takes care of the high-dimensional optimization, while BO incorporates human feedback.
> > Clarify the issue in lines 154-158...
>
> We apologize for the confusion. The problem with existing phosphene models (besides inferior performance) is that they are too slow for gradient descent and too large for GPUs. We also observed that training over large ranges of patients with previous models was unstable and did not converge. Lines 154-158 are some potential reasons for this, but ultimately it is an open area of research (effects of nonstandard NN functions on gradient loss landscape). Since we are uncertain in this regard, we will remove these lines.
> > Line 232 first mention of Argus II...
>
> We apologize for the omission, and have added a citation and explanation. Argus II is the most successful commercial visual prosthesis, with ~300 users [3], and was used to collect the data used for Table 1.
> > What does it mean to outperform the true DSE?
>
> For patients whose perception is precisely predicted by our phosphene model, the DSE with true $\phi$ is likely the ideal encoder for HILO. In reality no phosphene model is perfect, so it’s important for encoders to work when the assumed model is misspecified. For these patients, the true DSE is no longer guaranteed to be optimal, because the DSE was trained on non-misspecified patients. Our revised Fig 4 better illustrates that the True DSE performs worse on misspecified patients, while HILO performance is not as affected. HILO allows for users to guide optimization towards encoders that perform well for their misspecified forward models, a strong indicator that HILO is robust and has the potential to work on real patients.
> > How does the approach fail?
>
> For 1 of the 100 patients, HILO led to a final performance that was between the mean and random DSE baselines, but still an improvement over the naive encoder used by Argus II.

---

> > ### Comment · Reviewer_HJZJ · 2023-08-12
> >
> > Thank you for the additional detailed explanations. If the authors include their revised figures and additional details in the text, I think the paper will be greatly strengthened. I have read the other reviews and rebuttals, in particular the concerns raised by reviewer YFQD. Including the authors responses to clarify the application-driven nature of their work (vs new theory in ML) and to clarify some technical details on retinal prostheses, would make the paper much better, and I would raise my score to a 6.

---

> > > ### Author Response · Authors · 2023-08-13
> > >
> > > Thank you for the response. We are happy to include details requested by the reviewers in the revised/final version (open review doesn’t allow a revised version now, only the final version if accepted). This will include the updated figures, details on retinal prostheses and clarification of the application emphasis of the paper.

---

### Official Review · Reviewer_bw3i · 2023-07-06

**Soundness:** 3 good
**Presentation:** 3 good
**Contribution:** 3 good
**Rating:** 6
**Confidence:** 5

**Summary:**

In this work, the authors proposed a pipeline for optimizing the deep neural network based encoder, which is to generate visual stimulus for neuroprostheses. The pipeline considers human-in-the-loop optimization. The authors use preferential Bayesian optimization techniques to reduce the number of queries to the human making the decision. The major contributions claimed by the authors are as follows:
1. Introduced a forward model for retinal implants suitable for fast simulation experiments.
2. Proposed a personalized human-in-the-loop (HILO) patient-specific stimulus parameter optimization strategy using preferential Bayesian optimization.
3. Demonstrated HILO can quickly learn personalized stimulus encoder and the robustness of HILO, on simulated patients.


**Strengths:**

Originality and significance:
As far as I know, this work is the first work in considering the patient-specific stimulation parameter calibration problem for visual neuroprostheses that does not assume direct access to the output of the forward model. Instead, the authors introduce the technique of human-in-the-loop training with preferential Bayesian optimization in the pipeline. This setting is steps more realistic than previous works.

Quality and clarity:
The writing in general is very structured and easy to follow.


**Weaknesses:**

1. It is not made super clear whether and how the parameters w of the encoder is being updated in the manuscript.
2. The authors could have demonstrated a small amount of humans with sight to show the effectiveness of this optimization method.


**Questions:**

1. The simulated patient has access to t, which is the target percept or ground truth. Is it realistic to assume the blind person could have access to that?
2. In line 219 it’s written that the metric in equation (7) is also used to train the deep stimulus encoder (DSE). So the encoder’s parameters are getting updated in a rather conventional way as in the visual prosthesis literature. How to distinguish the contribution between the improvement of the encoder and the selection of patient-specific parameters? Could it be that the update of the encoder contribute the most? Ablation studies might be needed.


**Limitations:**

Other than the concerns listed in the weakness and question sections, the limitations mentioned in the work are adequate.

---

> ### Author Rebuttal · Authors · 2023-08-10
>
> We are grateful to the reviewer for their encouraging comments and thoughtful questions.
> ## Weaknesses
>
> > 1. It is not made super clear whether and how the parameters w of the encoder is being updated in the manuscript.
>
> We apologize for any confusion, and will update the paper with improved notations and clarity in this regard. During DSE training, the weights $w$ of the encoder are updated using standard gradient descent with the Adam optimizer. The loss function minimized during training is the reconstruction error between target images and the predicted percepts resulting from stimulation (Eq 7). Thus, the encoder learns to output optimal stimuli for given target images for the input patient specified by $\phi$. This all occurs before HILO begins. During HILO, the weights of the encoder are held constant, and are not updated. The only parameters optimized during HILO are the patient-specific parameters $\phi$, using the PBO strategy described in the paper.
>
>
> > 2. The authors could have demonstrated a small amount of humans with sight to show the effectiveness of this optimization method.
>
> We agree that this would have been an ideal way to verify whether our simulated subjects make decisions in the same way humans do. Unfortunately a human subjects experiment was outside the time and scope of this project. However, in [30], authors also conducted PBO on sighted humans in a similar, albeit much less realistic setting. Despite using a linear decoder and unrealistic phosphene model (as opposed to our deep neural network encoder and realistic phosphene model), their results still showed that PBO could be used with sighted humans viewing simulated prosthetic vision.
>
> ## Questions
>
> > 1. The simulated patient has access to t, which is the target percept or ground truth. Is it realistic to assume the blind person could have access to that?
>
> The reviewer raises an excellent point. With human patients, the subject indeed could not directly observe the target t. However, the user could be informed indirectly of the target, either verbally (e.g. asking ‘which looks more like the number 5’) and/or with tactile targets the subject could feel. In [30], authors demonstrated this with human subjects, who were only verbally told the target image.
>
> We do note that communicating targets with verbal or tactile representations would be limited to targets simple enough to communicate to the patient during training, such as letters, numbers, or simple objects. However, perception of simple stimuli would still be a huge improvement for current devices, which currently barely support letter recognition without prolonged head scanning [63]. Further, prior work ([12, 14, 36]) demonstrated that DSE’s also improve performance for natural images. We leave it as a question for future research, to determine how encoder improvements from HILO with simple stimuli transfer to more complex visual scenes.
>
> > 2. In line 219 it’s written that the metric in equation (7) is also used to train the deep stimulus encoder (DSE). So the encoder’s parameters are getting updated in a rather conventional way as in the visual prosthesis literature. How to distinguish the contribution between the improvement of the encoder and the selection of patient-specific parameters? Could it be that the update of the encoder contribute the most? Ablation studies might be needed.
>
> The reviewer is correct that the metric in equation (7) is used to train the DSE prior to HILO. During HILO, however, the DSE parameters are held constant, and only the patient-specific parameters $\phi$ are modified. Therefore, all of the performance improvements shown in the figures are due to the selection of patient-specific parameters.
> The other DSE baselines (marked as mean and random in the submitted Figure 3 and 4) are meant to serve as a sort of ablation, illustrating what performance would be like if only the DSE were used, without updating the patient-specific parameters.

---

> > ### Comment · Reviewer_bw3i · 2023-08-15
> >
> > Thank the author for the rebuttal. It addresses my questions 1 and 2. Could you elaborate more on the difference between [30] and improvement over it?

---

> > > ### Author Response · Authors · 2023-08-16
> > >
> > > In [30], authors used PBO to update a linear stimulus encoder based on feedback from sighted humans viewing simulated prosthetic vision. Each participant was shown the target image, encoded with 2 different linear encoders and fed back through a linear phosphene model, and chose which one they preferred. They used a linear approximation to [8] as their forward model (which does not match data from patients, see revised table 2), and a linear matrix inverse of this as their encoder. More generally, their approach is limited to linear forward models and stimulus encoders, and is fundamentally incompatible with more complex systems.
> > >
> > > Meanwhile, there is an increasingly large body of literature demonstrating that visual perception is a highly nonlinear function of electrical stimulation ([8,10,11,18,19,37,62], among others). Stimulation strategies that rely on assumptions of linearity have been ineffective at restoring high-resolution prosthetic vision in human subjects [3], generally reaching visual acuities much worse than theoretic limits based on device hardware [Stronks and Dagnelie 2013].
> > >
> > > Our framework for integrating PBO with a deep stimulus encoder is the first approach that does not require the forward model or stimulus encoder to be linear. This not only led to large improvements, both in terms of realism and performance, for visual prostheses, but also has the potential for much broader applicability. Our HILO strategy could work for any system that can be inverted using a DNN, potentially enabling its use for other modalities outside of prosthetic vision.
> > >
> > >
> > > Other smaller differences:
> > > - Our DNN is trained using a perceptual metric (Eq 7). Their encoder is a linear matrix inverse and thus only minimizes MSE, which has been shown to be misaligned with human visual perception [Lin and Kuo, 2011].
> > > - Since we used simulated patients, we were able to test more patients spanning a wider, more realistic range.
> > > - We conducted additional experiments demonstrating the robustness of our approach.

---

> > > > ### Comment · Reviewer_bw3i · 2023-08-16
> > > >
> > > > Many thanks for the clarification. Please include that in the revised version in the related work section or discussion.
> > > >
> > > > This work is a nice combination of available methodologies toward more realistic visual neuroprosthetics. The major contribution is on the calibration side since humans/animals cannot be probed in the same manner as deep learning models (no 'gradient', limited numbers of queries). Thus, such calibration techniques are worth studying considering their potential great future impact when other technologies of neuroprosthetics became mature enough. I will keep the current rating but raise my confidence from 3 to 5.

---

### Official Review · Reviewer_YFQD · 2023-07-24

**Soundness:** 3 good
**Presentation:** 2 fair
**Contribution:** 3 good
**Rating:** 5
**Confidence:** 3

**Summary:**

* This paper proposes a flexible framework addressing personalized stimulus optimization predominantly seen in _visual prostheses_.
* The authors propose integrating the state-of-the-art deep learning with a preferential Bayesian Optimization (BO) strategy to learn optimal patient-specific parameters in fewer trials.
* The proposed approach is a two-stage process, with the first stage aiming at learning a Deep Stimulus Encoder (DSE) to optimize stimuli, and the second stage aiming at embedding the DSE learnt in the first stage in a sample-efficient preferential Bayesian optimization strategy.

**Strengths:**

* This paper aims at advancing the state-of-the-art in sensory neuroprostheses that have a significant impact on the present world.
* The authors propose to overcome the strong assumptions made on the accuracy of the knowledge on patient-specific parameters, thus aiming to improvise the accuracy of the deep stimulus encoder by optimizing the stimuli.
* The authors use a sample-efficient Bayesian optimisation (Human-in-the-loop Optimization (HILO)) method to optimally select the patient-specific parameters used in deep stimulus encoder.
* Authors have empirically evaluated the proposed approach in optimizing the personalized stimulus encoder for several simulated patients.
Further, the authors evaluate the performance of their method when the preferential inputs are noisy, thereby proving its robustness.


**Weaknesses:**

* The main weakness of this paper lies in its _limited novelty_, as the proposed approach is merely a _combination_ of the state-of-the-art of forward models and sample-efficient Bayesian optimization to optimally find personalized stimuli.
* Although this paper improvises the techniques used in the visual prostheses domain, the overall contribution of this paper is in the _application of the existing ML/DL techniques_ in neuroprostheses, visual prostheses in particular, thus may not advance the state-of-the-art in ML/DL community.
* Though the paper reads well in many parts, it demands significant knowledge/domain expertise in visual prosthesis for a thorough understanding of the paper.
* Authors have missed defining/describing a few important mathematical quantities or parameters that break the flow of understanding and also raise questions about its reproducibility (Please refer to the Questions section).
* The details provided in _Section 4_ give the impression that the proposed framework is tightly coupled with the problems in visual prostheses.
* The authors have discussed a few experimental settings in Section 4 that are apt to discuss in Section 5, though they do not contribute to the proposed method.
* The visualization schemes used to show the empirical results do not effectively demonstrate the superiority of the proposed approach. Also, the authors do not discuss in brief the trends and the possible reasons behind them.


**Questions:**

* Line 159: What does $n_e$ stand for? number of electrodes?
* Line 169: What is the significance of threshold $\epsilon$ and why authors have set it to $e^{-2}$?
* Line 198: Why do authors considered only 10 simulated patients for transfer learning? is that not sparse?
* Line 200: The hyperparameters considered and their bounds are neither mentioned in the main paper nor in the supplementary material. Could authors confirm the hyperparameter set of each kernel?
* Line 214: The noise parameter s is set to be $\frac{1}{0.01}=100$, whereas in Line 285, s is set to 1e-4, is this a typo or it is $\frac{1}{1e-4}$?
* Line 221: What is the significance of $\beta$ in equation (7)?
* Line 235: What does $R_i$ stands for? Is $i \in$ {$ 1,...,n_e$}?
* Line 239: Why authors have not considered comparing their empirical results with Beyeler et al. 2019?
* Line 250: How do the results compare with Granley et al. 2022 if L1 is used in computing the test loss for DSE?
* Line 302: Could Batch-BO strategies further reduce the latency and thus provide a quicker solution by parallelizing the data acquisition?
* Figure 3: Why there is an increasing (or decreasing) trend in loss (or accuracy) plot during the initial phase of the optimization?
* Figure 4: Why there is a lot of variance in the bottom right plot for "Out of Distribution" misspecification?


Following are the minor (typographical) errors, that needs to be fixed.

* Inconsistent referencing style and hyperlinks are missing throughout the paper. It is strongly suggested to not modify the style files provided by NeurIPS.
* Line 59: "...visual prostheses that, where ...."
* Figure 2: The current font size in the Leftmost plot is illegible.
* Line 151: The section/Sub-section number has to be clearly mentioned.
* Line 204: Provide a quick reference to the Brier score in the main paper, even though it is mentioned in the supplementary.
* Line 237: "ratings as amplitude, frequency,...."
* Line 259: :...  perceptual loss (figure 3.B)....(figure 3.C)"

**References**

* Michael Beyeler, Devyani Nanduri, James D. Weiland, Ariel Rokem, Geoffrey M. Boynton, and Ione Fine. A model of ganglion axon pathways accounts for percepts elicited by retinal implants. Scientific Reports, 9(1):1–16, June 2019.

**Limitations:**

* The optimization performance of PBO crucially depends on the _ generalization performance of the inherent GP surrogate model_. Authors reduce latency by avoiding online inference and instead use a transfer learning strategy. However, the source considered in the adopted transfer learning uses sparse data generated from 10 patients, thus, raising concerns about the optimality of kernel parameters and generalizability of the GP surrogates.
* Although the authors discuss the empirical results, the lack of sufficient experiments and competing baselines (Beyeler et al. 2019, Granley et al. 2022) in the empirical results makes it hard to conclude the superiority of the proposed approach. (Please refer to the Questions section).
* Sequential PBO may not be very suitable in time-critical situations, thus, more efficient strategies such as batch-BO could be of great use to reduce the possible latency in finding the optimal stimuli on-the-fly.

**References**

* Michael Beyeler, Devyani Nanduri, James D. Weiland, Ariel Rokem, Geoffrey M. Boynton, and Ione Fine. A model of ganglion axon pathways accounts for percepts elicited by retinal implants. Scientific Reports, 9(1):1–16, June 2019.

* Jacob Granley, Lucas Relic, and Michael Beyeler. Hybrid Neural Autoencoders for Stimulus Encoding in Visual and Other Sensory Neuroprostheses. October 2022.

---

> ### Author Rebuttal · Authors · 2023-08-10
>
> We are grateful for the reviewer’s attentive analysis and helpful feedback.
> ## Weaknesses/Limitations
> > The main weakness lies in its limited novelty, as the proposed approach is merely a combination of state-of-the-art forward models and Bayesian optimization.
>
> As the reviewer points out, our work builds on several previous ideas. However, combining these ideas into a realistic HILO scheme required significant improvements to each individual component, as well as complex design choices to integrate components into the optimization framework. Our specific points of novelty are:
> - A HILO strategy combining PBO with deep stimulus encoding. Our DSE was trained to output optimal stimuli conditional upon a latent set of patient-specific parameters, allowing PBO to optimize the latent parameters instead of the high dimensional stimuli. This unique combination addresses the main limitations of previous approaches to HILO in BCI: reliance on simplistic encoders [30] or very limited optimization dimensionality [55-61]. HILO proved highly effective, improving phosphene quality and robustness (Figures 3, 4).
> - A new phosphene model, which matches patient data significantly better (revised Table 2) and improves computational efficiency (~50x faster, ~120x less memory than [14]).
> - A significantly improved DSE, which performed better than the SOTA (.108 vs .12 L1 loss) [14], despite having realistic (13 vs 2) parameters of patient variations. Previous approaches require retraining for new patients, preventing HILO.
>
> > This paper … may not advance state-of-the-art in ML/DL
>
> We agree advances in this paper mainly benefit the neuroprosthesis and BCI communities. Thus, this paper targets the “Neuroscience and cognitive science (e.g., neural coding, brain-computer interfaces)” track in the call for papers.
> > ...the proposed framework is tightly coupled with visual prostheses.
>
> While our implementation is specific to visual prostheses, the proposed framework of 1) building a forward model, 2) training a DSE across patients and 3) learning optimal patient parameters is not domain specific. Forward models [48-51] and DSEs [52-54] have been developed for multiple sensory modalities and could potentially be adapted for HILO.
> > GP Hyperparameter Optimality
>
> The reviewer raises an excellent point. We consider it a strength that HILO worked using only 10 transfer learning patients, which was chosen to match clinical settings with limited human data availability. To test optimality we ran HILO again, finding hyperparameters for each patient prior to optimization using 200 random duels. Using these ‘patient-optimal’ hyperparameters, HILO performed similar to the transfer approach (Fig R3). While keeping hyperparameters constant is common [57-60], online optimization is an alternative method. We tested this using update periods of 1, 5, 10, or 20 duels (Fig. R1). All update periods eventually converged to similar performance as with transfer learning, but required more optimization time. Together, these suggest that transfer learning is a good strategy for HILO.
> > Comparing results with Beyeler 2019 [8]
>
> Although the previous SOTA model [11] is a simple improvement to [8], we agree that readers unfamiliar with these details would benefit from a direct comparison and have added additional baselines to Table 2 (see pdf).
> > Batch BO
>
> While sequential BO would be quick (~17 minutes with 100 iterations, 10 seconds each), batch-BO indeed could speed up optimization. We consider 2 options: 1) the user compares a batch of stimuli in each iteration, and 2) batches of duels are precomputed, to save on acquisition time between duels. While 1) acquires more information per response, patients must remember multiple stimuli before making comparisons, increasing cognitive burden. Future work might consider whether parallelization of data acquisition makes up for the increased difficulty. We tested 2) using batched MUC [40] with 1, 3, 6, or 10 duels (Fig R3). Batched MUC reduced acquisition time required to reach a desired performance, at the cost of requiring more duels. In practice, batch size could be determined by balancing patient response and acquisition time. Faster acquisitions (e.g. KernelSelfSparring [64]) could further improve this time.
> ## Questions
> > What does $n_e$ stand for?
>
> Number of electrodes
> > What is the significance of $\epsilon$, why $e^{−2}$?
>
> The phosphene model is derived so an electrode’s phosphene will have area (#pixels > $\epsilon$) of $\rho$. $e^{-2}$ was chosen based on thresholds in [11].
> > Could authors confirm the hyperparameter set?
>
> For lack of space, we refer to [4] supplementary 4, who used the same kernels and hyperparameters (lengthscales and scaling factor), each with bounds of [exp(-10), exp(10)].
> > s is set to be 1/0.01, whereas in Line 285, s is set to 1e-4, is this a typo?
>
> Yes, the referenced quantity should be 1/1e-4. We will modify Eq 6 and update notation through the paper.
> > What is the significance of $\beta$ in Eq 7?
>
> $\beta$ controls the weighting between MSE and VGG extracted features. See [14] App B for discussion.
> > What is $R_i$?
>
> $R_i^2$ is the coefficient of determination ($R^2$) for the ith shape descriptor (area, eccentricity, orientation). See equations 10-13 in [8] for more details.
> > Why is there an increasing trend in loss during the initial phase of optimization?
>
> This is a common pattern in PBO (e.g. Fig. 5 in [22]). The estimate of the maximum of the GP mean (Eq. 4) rapidly changes in initial phases of optimization, meaning the optimum estimate might worsen. This is in contrast to noise-free approaches, where the maximum is taken over previously sampled points and guaranteed to never worsen performance.
> > Why there is a lot of variance for OOD misspecification?
>
> Note the smaller y axis on the OOD plot, causing variance to appear larger. We have improved the visualization to use uniform scaling, showing similar variance for OOD patients (Revised Fig 4).

---

> > ### Comment · Reviewer_YFQD · 2023-08-13
> >
> > I would like to thank the authors for their efforts in providing a detailed rebuttal. I have gone through the other reviews and the corresponding rebuttals and I expect the authors to incorporate the changes to improvise the quality of the paper. The authors have sufficiently addressed my concerns and thus I am slightly increasing my scores.

---

### Author Rebuttal · Authors · 2023-08-10

We thank the reviewers for their thoughtful analysis and feedback, which has been invaluable for understanding how to improve our paper. We are pleased that reviewers in general agreed on the paper’s significance towards realistic optimization of prosthetic vision, and that it advances state of the art in this domain (YFQD, bw3i, HJZJ). We are also glad reviewers found HILO to be “steps more realistic than previous approaches” (bw3i), well evaluated (HJZJ), and robust (YFQD). While general ideas behind individual components of our algorithm have been proposed in isolation (phosphene model, deep stimulus encoding, PBO), most reviewers recognized the novelty and practical feasibility of our approach, which improves each component while integrating them into a realistic and efficient optimization pipeline (bw3i and HJZJ). Reviewers raised thoughtful questions about Batch BO and BO hyperparameters (YFQD), use of simulated data (HJZJ), and the feasibility of the approach on human patients (bw3i, HJZJ).

In the following responses, we address each of the reviewers’ questions and concerns in detail. Additional experiments were conducted regarding batch Bayesian optimization and optimization of GP kernel hyperparameters to address reviewer’s feedback. We have uploaded a pdf containing figures for these experiments, a revised Table 2 with more baseline comparisons, as well as improvements in visualization schemes for original figures 3 and 4, in accordance with reviewer feedback. Reviewer feedback will be integrated into a revised version of the paper, which, if accepted, will present our results with improved clarity and include the experiments and improvements suggested by reviewers.

## Additional References for rebuttals:

[46] Luo et al. 2016. “Long-Term Repeatability and Reproducibility of Phosphene Characteristics in Chronically Implanted Argus II Retinal Prosthesis Subjects.” American Journal of Ophthalmology

[47] Yue et al. 2021. “Restoring Color Perception to the Blind – an Electrical Stimulation Strategy of Retina in Patients with End-Stage Retinitis Pigmentosa.” Ophthalmology

[48] Dorman et al. 2005. “Acoustic Simulations of Combined Electric and Acoustic Hearing (EAS).” Ear and Hearing

[49] Okorokova et al. 2018. “Biomimetic Encoding Model for Restoring Touch in Bionic Hands through a Nerve Interface.” Journal of Neural Engineering

[50] Saal et al. 2017. “Simulating Tactile Signals from the Whole Hand with Millisecond Precision.” Proceedings of the National Academy of Sciences

[51] Mileusnic et al. 2006. “Mathematical Models of Proprioceptors. I. Control and Transduction in the Muscle Spindle.” Journal of Neurophysiology

[52] Drakopoulos et al. 2023. “A DNN-Based Hearing-Aid Strategy For Real-Time Processing: One Size Fits All.” In ICASSP 2023 - 2023 IEEE International Conference on Acoustics, Speech and Signal Processing (ICASSP)

[53] Drakopoulos et al. 2022. “A Differentiable Optimisation Framework for The Design of Individualised DNN-Based Hearing-Aid Strategies.” In ICASSP 2022 - 2022 IEEE International Conference on Acoustics, Speech and Signal Processing (ICASSP)

[54] Drakopoulos et al. 2023. “A Neural-Network Framework for the Design of Individualised Hearing-Loss Compensation.” IEEE/ACM Transactions on Audio, Speech, and Language Processing

[55] Ding et al. 2018. “Human-in-the-Loop Optimization of Hip Assistance with a Soft Exosuit during Walking.” Science Robotics

[56] Nielsen et al. 2015. “Perception-Based Personalization of Hearing Aids Using Gaussian Processes and Active Learning.” IEEE/ACM Transactions on Audio, Speech, and Language Processing

[57] Louie et al. 2021. “Semi-Automated Approaches to Optimize Deep Brain Stimulation Parameters in Parkinson’s Disease.” Journal of NeuroEngineering and Rehabilitation

[58] Lorenz et al. 2019. “Efficiently Searching through Large TACS Parameter Spaces Using Closed-Loop Bayesian Optimization.” Brain Stimulation

[59] Tucker et al. 2020. “Preference-Based Learning for Exoskeleton Gait Optimization.”

[60] Lorenz et al. 2021. A Bayesian optimization approach for rapidly mapping residual network function in stroke. Brain 144, 2120–2134

[61] Zhao et al. 2021. “Optimization of Spinal Cord Stimulation Using Bayesian Preference Learning and Its Validation.” IEEE Transactions on Neural Systems and Rehabilitation Engineering

[62] Nanduri, Devyani. 2011. “Prosthetic Vision in Blind Human Patients: Predicting the Percepts of Epiretinal Stimulation.” ProQuest Dissertations and Theses. Ph.D., United States -- California: University of Southern California.

[63] Da Cruz et al. 2013. “The Argus II Epiretinal Prosthesis System Allows Letter and Word Reading and Long-Term Function in Patients with Profound Vision Loss.” British Journal of Ophthalmology

[64] Sui, Yanan, Vincent Zhuang, Joel W. Burdick, and Yisong Yue. "Multi-dueling bandits with dependent arms." arXiv preprint (2017).

---

### Decision · Program_Chairs · 2023-09-21

**Decision:**

Accept (poster)

**Comment:**

This paper discusses a method to personalize stimulation parameters for a visual prosthesis.
All reviewers were in favor of publication.   The results are impressive, but the weakest part of the
paper is that the studies are done on "simulated people".   There was a robust rebuttal discussion
and reviewers were happy with the outcome but request that these new details be added to the paper.